METHODS

# iSTTC: A robust method for accurate estimation of intrinsic neural timescales from single-unit recordings

Irina Pochinok[1]*, Ileana L. Hanganu-Opatz[1], Mattia Chini[1,2]*

**1** Institute of Developmental Neurophysiology, Center for Molecular Neurobiology, University Medical Center Hamburg-Eppendorf, Hamburg, Germany, **2** GIGA Neurosciences, University of Liège, CHU Sart Tilman, Liège, Belgium

* irina.pochinok@zmnh.uni-hamburg.de (IP); mchini@uliege.be (MC)

## Abstract

Intrinsic neural timescales (ITs) are an emerging measure of how neural circuits integrate information over time. ITs are dynamically regulated by behavioral context and cognitive demands, making them suitable for mapping high-level cognitive phenomena onto the underlying neural computations. In particular, IT measurements derived from single-unit activity (SUA) offer fine-grained resolution, critical for mechanistically linking individual neuron dynamics to cognition. However, current methods for estimating ITs from SUA suffer significant biases and instabilities, particularly when applied to sparse, noisy, or epoched neural spike data. Here, we introduce the intrinsic Spike Time Tiling Coefficient (iSTTC), a novel metric specifically developed to address these limitations. Leveraging synthetic and experimental single-unit recordings, we systematically assessed the performance of iSTTC relative to traditional approaches. Our findings demonstrate that iSTTC provides more accurate IT estimates across a wide range of conditions, reducing estimation error especially in challenging yet biologically relevant regimes. Crucially, iSTTC can be applied to both unsegmented and epoched data, overcoming a critical limitation of existing methods. Furthermore, iSTTC substantially relaxes inclusion criteria, increasing the fraction of neurons suitable for analysis and thereby improving the representativeness and robustness of IT measurements. The methodological advances introduced by iSTTC represent a substantial step forward in accurately capturing neural circuit dynamics, ultimately enhancing our ability to link neural mechanisms to cognitive phenomena.

## Author summary

Undestanding how the brain integrates information over different timescales is an important question in neuroscience. Intrinsic neural timescales, derived from the autocorrelation structure of neural activity, provide a window into these

**Data availability statement:** Synthetic dataset is available at https://gin.g-node.org/iinnpp/isttc. Experimental data were obtained from https://allensdk.readthedocs.io/en/latest/visual_coding_neuropixels.html. Analysis and visualization code is available at https://github.com/iinnpp/isttc.

**Funding:** German Research Foundation FOR5159 TP1 (437610067) to I.L.H.-O. (www.dfg.de) and the Fonds de la Recherche Scientifique - FNRS - MISU (40032471) to M.C. Funders did not play a role in the study design, data collection and analysis, decision to publish, or preparation of the manuscript.

**Competing interests:** The authors have declared that no competing interests exist.

integration processes. They vary with behavioral context and cognitive demands, and single-unit recordings offer the most precise way to examine them. However, existing estimation methods can be biased, unstable, or overly restrictive. In this study, we introduce iSTTC, a new method designed to measure intrinsic timescales more accurately and more reliably from single-neuron recordings. Using both simulated and real neural data, we show that iSTTC performs better than commonly used approaches, works well on both continuous and trial-based recordings, and allows many more neurons to be included in the analysis. This means that we can obtain more representative and robust measurements of neural dynamics, even under challenging conditions. By improving how intrinsic timescales are estimated, our method helps pave the way toward a deeper understanding of how neural circuits process information across time.

## 1 Introduction

A remarkable ability of our brains is that of being capable of choosing an appropriate course of action over extremely different timescales: from a rapid millisecond-range reaction to avoid an unforeseen obstacle while driving a car, to choosing the next move in a complex chess match, which might unfold over several minutes. Intrinsic neural timescales (ITs) are an emerging fundamental measure quantifying how different systems integrate information over time [1,2]. In this manuscript, we define ITs specifically as the exponential decay constant of the spike-count autocorrelation of single-unit activity; for a broader discussion of alternative timescale metrics and their interpretation across signals and methods, see [3]. ITs vary significantly across the brain [1], which is thought of reflecting the computational properties of different brain regions. Within the neocortex, regions that sit at the bottom of the cortical hierarchy, such as primary sensory cortices, typically exhibit short timescales, conducive to rapid sensory processing. Higher-order brain areas, such as the prefrontal cortex, display substantially longer timescales, better suited to supporting more complex cognitive functions [1,4]. Remarkably, these findings have been robustly replicated across phylogenetically distant species and disparate recording techniques, including single-unit activity (SUA), 2-photon and widefield calcium imaging, local field potentials (LFP), electrocorticography (ECoG), electro- and magneto-encephalography (EEG and MEG), and functional magnetic resonance imaging (fMRI) [2,4–15].

While few experimental studies have mechanistically investigated the factors that underlie this IT heterogeneity, early indications point to several single-neuron and network-level factors as being potentially implicated [4,16–19]. For instance, higher proportions of somatostatin-expressing interneurons and prolonged NMDA currents positively correlate with longer ITs, whereas parvalbumin-expressing interneurons, high firing rates and large dendritic arborizations generally contribute to shorter ITs [4,17,20]. The synaptic, cellular and network mechanisms underlying the properties of ITs are reviewed in depth elsewhere [3,21–23]. Importantly, ITs are not fixed but dynamically modulated by behavioral context, cognitive demands, and

task engagement [4,7,10,11,24]. Another important property of ITs lies in their predictive power for behavioral and cognitive performance. For example, shorter ITs correlate with faster reaction times [25]. This suggests that modulation of ITs could be a mechanism by which neural circuits dynamically adjust computational resources to behavioral needs. Thus, a detailed understanding of the mechanisms regulating ITs might represent a promising approach to bridge neural dynamics with higher-order cognitive functions, and could even serve as mechanistic biomarkers in clinical contexts [26–28]. To this aim, recording modalities that bridge single neuron and network levels of investigations, such as SUA and 2-photon calcium imaging, are particularly promising for advancing our mechanistic understanding of ITs.

Despite their potential importance, current methods for estimating ITs from SUA exhibit substantial limitations [3]. Typically, IT estimation involves a two-step process [3]:

1. calculating an autocorrelation (or autocorrelation-like) function (ACF, but see below);

2. fitting an exponential decay function.

Of note, in this paper we reserve the use of "ACF" for the classically defined autocorrelation function. Any alternative pre-processing, smoothing, or surrogate-based approximations are referred to as "ACF-like" methods. Recent methodological improvements have primarily focused on optimizing the exponential fitting step, with approaches such as adaptive Approximate Bayesian Computation (aABC) substantially reducing bias and improving robustness [29]. However, significant uncertainty remains in the initial process of estimating the ACF itself. This is a critical step whose results are influenced by multiple factors, including sensitivity to data length [30], the SUA firing rate, the eventual presence of trials, the representation of the spiking data as binary or continuous, the choice of bin width or smoothing kernel, and other preprocessing choices. Furthermore, most existing trial-based methods impose strict inclusion criteria, thus restricting analysis to a minority of recorded neurons [1,8,31–34]. This runs counterintuitive to the notion that ITs are based on the decay rate of all neurons in a certain brain area. Additionally, given that the ACF cannot be readily estimated on epoched, non-continuous data, the most commonly used approach for trial-based IT estimation (referred to as PearsonR in this manuscript, see Materials and Methods for details) does not calculate the genuine ACF, but instead measures pseudo-autocorrelations on trial-averaged data [1].

To address these limitations, we introduce a novel metric: the intrinsic Spike Time Tiling Coefficient (iSTTC). iSTTC, an extension of STTC [35], robustly estimates ITs and offers significant improvements over traditional ACF-based approaches. It provides more accurate IT estimates on long, uninterrupted spiking data, particularly under biologically realistic cortical conditions characterized by low firing rates and high recurrent connectivity. Unlike traditional ACF methods, iSTTC is unaffected by zero-padding, and thus also seamlessly generalizes to epoched spiking data after 0-padding and concatenation. In simulated trial-based data, iSTTC consistently outperforms the PearsonR method, producing more accurate estimates of the true underlying timescales, particularly if the number of trials is low. Lastly, iSTTC relaxes the stringent inclusion criteria required by other commonly used trial-based methods, thus enabling IT estimation in a substantially larger fraction of neurons. This broader applicability enhances both the statistical power and representativeness of the estimated ITs. iSTTC thus enhances the reliability of the estimation of the "ACF-like", the first step of IT measurements, analogously to what methods such as aABC have done for the exponential fitting step, the second part of the process.

In summary, iSTTC is a robust, accurate, and broadly applicable tool for IT estimation from SUA signals. By improving the reliability of IT measurements, iSTTC advances our ability to link intrinsic neural dynamics to functional and behavioral outcomes, strengthening ITs' potential role as mechanistic biomarkers for cognitive processing.

## 2 Results

### 2.1 iSTTC calculates an autocorrelation-like function on non-binned spike trains

ITs are defined as the decay time constant of the signal's autocorrelation function (Fig 1A). Estimation of ITs is a two-step process (Fig 1B): first, computing the ACF (or ACF-like), and second, estimating its decay time constant.



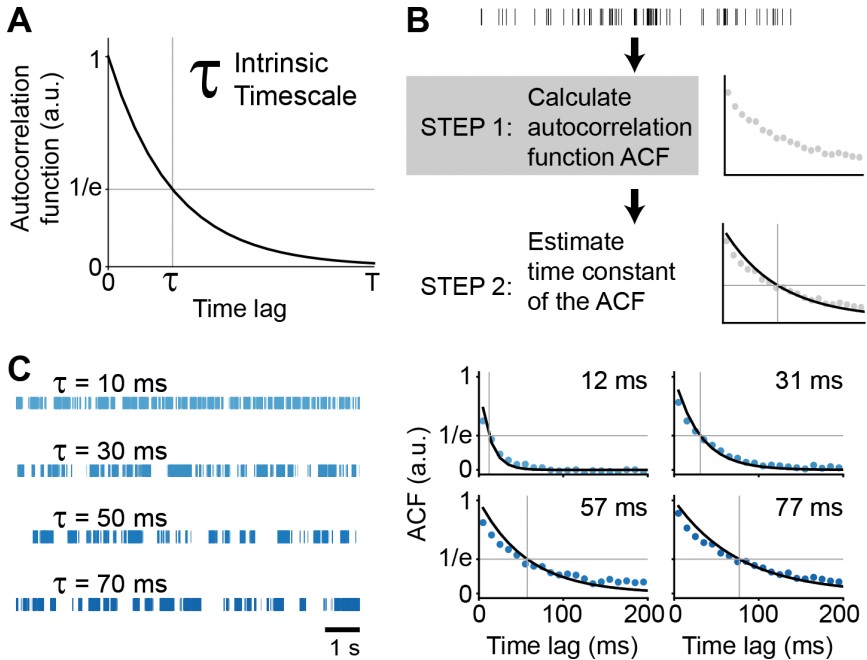

**Fig 1. Definition and schematic representation of intrinsic timescale estimation. (A)** Schematic illustration of an intrinsic timescale, defined as the decay time constant of the spike train ACF. **(B)** Schematic of the intrinsic timescale estimation pipeline for single-unit activity. **(C)** Representative examples of spike trains with known and increasing from top to bottom intrinsic timescales (left) and their corresponding estimated intrinsic timescales (right). In **(C)**, blue dots represent the computed ACF, and the black line indicates the fitted exponential decay function.

Methods for estimating decay time constant based on the ACF/ACF-like can be broadly categorized into model-free and model-based approaches [3]. In this paper we focus on optimizing the first step (the calculation of the ACF/ACF-like) and use a model-based approach that assumes that the ACF/ACF-like follows an exponential decay (Fig 1C).

The calculation of the ACF/ACF-like, although seemingly straightforward, can be influenced by several arbitrary choices. First, the time series used to estimate the ITs can differ in structure: they may consist of long, continuous, unsegmented spike trains [36] or a set of shorter trials, typically taken from baseline or fixation periods within behavioral tasks when the subject is not actively engaged [1,31]. Second, spiking data can be represented either as binary spike times or as continuous signals obtained by binning or smoothing the spike train, which influences the method used to calculate the ACF/ACF-like. Both the structure of the data and the signal representation can introduce variability in the resulting ACF/ACF-like and, by extension, in the IT estimates.

To investigate the effects of these factors, we estimated ITs on a large synthetic dataset and an experimental one (Table 1).

To calculate the ACF/ACF-like, we used three different methods (Table 2). Two of these are commonly used in the literature: i) the classic ACF for unsegmented signals; ii) the Pearson's correlation coefficient on averaged signal (PearsonR) for epoched data. We compared these methods to iSTTC, the measure that we introduce in this manuscript (Fig 2B and 2C). iSTTC can be used as an ACF-like measure for both unsegmented and epoched data.

To compute an ACF-like of unsegmented signals using iSTTC, we used a modified version of STTC (Spike time tiling coefficient) [35]. The STTC is defined as follows: $STTC_{\Delta t} = \frac{1}{2}\left(\frac{P_A - T_B}{1 - P_A T_B} + \frac{P_B - T_A}{1 - P_B T_A}\right),$ (1)

**Table 1. Summary of used datasets.**

|  | Synthetic SUA | Mouse single unit activity |
|---|---|---|
| **Reference** | Datasets were generated in this study | [2] |
| **Dataset source** | gin.g-node.org/iinnpp/isttc | Openly available for download in Neurodata Without Borders format via the AllenSDK |
| **Dataset info** | $10^5$ single units with varying spike train features (firing rate, intrinsic timescale and burstiness) | 5775 single units, 26 mice, 8 brain areas, 30 min long spike trains and trials |
| **Related figures** | Fig 3, Fig 4, Fig 5, Fig 6 | Fig 7 |
| **Investigated biases** | Trial number, signal length, unsegmented vs epoched data, firing rate, IT magnitude, excitation strength | Trial number, signal length, unsegmented vs epoched data |

**Table 2. Summary of used methods.**

|  | Binary signal (spike times) | Binned spike train |
|---|---|---|
| **Unsegmented** signal | iSTTC | ACF |
| **Trials** | iSTTC (trials) | PearsonR |

where $STTC_{\Delta t}$ is a correlation between spike trains $A$ and $B$ on the time interval $\Delta t$; $T_A$ ($T_B$) is the proportion of total signal length that lies within $\pm\Delta t$ of any spike of $A$ ($B$); $P_A$ ($P_B$) is the proportion of spikes from $A$ ($B$) that lies within $\pm\Delta t$ of any spike of $B$ ($A$) (Fig 2A).

The STTC formula is ill-suited for computing the autocorrelation. By design, STTC centers each time lag on individual spikes and uses expanding intervals around those spikes rather than a sliding window of variable size. As a result, applying STTC to the same spike train always yields a correlation of 1, since each spike coincides with itself.

To use the STTC for autocorrelation, we compared the spike train to temporally shifted versions of itself. For each time lag $lag_k$, for the spike train A, we truncated the end of the spike train by removing spike times >= $signal\ length - lag_k$ and for the spike train B, we truncated the original spike train by removing the spike times < $lag_k$ so that both sequences remained the same length. We then realigned spike train B by subtracting the time lag $lag_k$ from its spike times so that both spike trains occupied the same time window. We then computed STTC between these two aligned spike trains. Thus, if, for instance, we want to compute the ACF-like in 50 ms bins, a 50 ms shift yields the first autocorrelation lag, 100 ms the second, and so on (Fig 2B). We defined the iSTTC autocorrelation at $k$ lag as follows:

$$iSTTC_{lag=k} = \frac{1}{2}\left(\frac{P_{k:T} - T_{1:T-k}}{1 - P_{k:T}\,T_{1:T-k}} + \frac{P_{1:T-k} - T_{k:T}}{1 - P_{1:T-k}\,T_{k:T}}\right)$$

(2)

For epoched data, before applying iSTTC, we linearize the signal by concatenating the trials. Because iSTTC compares spike timing within a continuous time window, directly concatenating trials would introduce spurious correlations across trial boundaries. To avoid these artefacts, we concatenate trials with a zero-padding, ensuring that spikes from adjacent trials do not contribute to the iSTTC calculation. The length of the zero padding must be equal to or greater than the length of the trial (Fig 2C).

For each time lag $lag_k$, we construct two sets of trial-wise spike segments: a shifted segment containing spikes occurring after $lag_k$ (realigned by subtracting $lag_k$), and an unshifted segment containing spikes occurring before



**Fig 2. Schematic representation of spike time tiling coefficient calculation (STTC) and its adaptation for intrinsic timescale estimation (iSTTC). (A)** Schematic representation of STTC calculation, modified from [35] and [37]. STTC quantifies the correlation between spike trains A and B at a time interval $\Delta t$. $T_A$ ($T_B$) denotes the proportion of the signal within $\pm \Delta t$ of any spike in A **(B)**, and $P_A$ ($P_B$) the proportion of spikes in A (B) that fall within $\pm \Delta t$ of a spike in B **(A)**. **(B)** Schematic illustration of iSTTC. Spike trains A and B are created by truncating and realigning the original spike train, after which the standard STTC formula is applied as in **(A)**. **(C)** Schematic illustration of iSTTC applied on epoched data. Each trial is zero-padded prior to computing iSTTC across lags. Bottom left, violin plot displaying the absolute difference in estimated intrinsic timescales relative to the reference condition in which the zero-padding length equals the trial length $T$.

*signal length* − $lag_k$. After concatenating the corresponding segments across trials with zero-padding, we compute the STTC between the two concatenated spike trains.

The STTC implementation for trials differs from the STTC on unsegmented data in how the $T$ term is calculated. Since $T$ is defined as the proportion of total signal length that lies within $\pm \Delta t$ of any spike, it depends on the length of the signal. Zero padding artificially increases the total signal duration and would therefore change $T$. To avoid this artefact, we compute $T$ within each original trial before padding, using the true trial duration, and then average these values across trials. This provides a fixed and unbiased $T$ for each time lag and ensures that changes in the autocorrelation across lags reflect genuine temporal structure rather than changes in the signal length introduced by padding. The calculation of the $P$ term is unaffected by the signal length. This implementation ensures that zero padding does not affect the iSTTC calculation.



## 2.2 iSTTC outperforms ACF in IT estimation accuracy on unsegmented spiking data

To compare the performance of iSTTC and ACF in estimating ITs on long, continuous, unsegmented spiking data, we used a self-exciting Hawkes point process to simulate $10^5$ spike trains of 10 minutes each (Fig 3A, left). The spike trains varied systematically in time constant of the Hawkes point process (i.e., the IT to estimate; from 50 to 300 ms), firing rate (from 0.01 to 10 Hz), and excitation strength (from 0.1 to 0.9; see Materials and Methods for details). Each parameter was uniformly sampled within the specified range (Fig 3A, right). To compare the IT estimation accuracy of the two methods, for each iteration we computed the Relative Estimation Error (REE), defined as $REE = \frac{IT_e - IT_{gt}}{IT_{gt}} \times 100$, where $IT_e$ is the estimated IT and $IT_{gt}$ is the ground truth IT (Fig 3B).

Across the overwhelming majority of the explored parameter space, iSTTC yielded lower REEs than ACF (Fig 3C). To rigorously quantify this difference, we modeled how the IT estimation method impacted REEs with a multivariate generalized linear model. Across the entirety of the synthetic dataset, we found a robust effect of IT estimation method (coefficient estimate = −0.037, 95% CI [-0.039, -0.034], p < $10^{-16}$; S2A Fig), indicating that the REE of iSTTC was, on average, 8% lower than that of ACF. This difference was not uniform throughout the parameter space (Fig 3C). Accordingly, statistical modeling revealed that the effect of IT estimation method significantly interacted with firing rate (coefficient estimate = 0.003, 95% CI [0.0007, 0.0061], p = 0.0124; Figs 3D, left and S2A), excitation strength (coefficient estimate = -0.037, 95% CI [-0.039, -0.034], p < $10^{-16}$; Figs 3D, middle and S2A) and the value of the IT (coefficient estimate = 0.005, 95% CI [0.003, 0.009], p < $10^{-6}$; Figs 3D, right and S2A). Thus, estimating ITs with iSTTC resulted in lower REEs than ACF, an advantage that was particularly evident in low-firing rate, high excitation strength and low IT values regimes (Figs 3C–3D and S1).

While firing rate and IT are parameters that can be evaluated also in experimental datasets, excitation strength is not. Thus, we set out to investigate whether we could identify an experimentally-observable proxy for excitation strength, and reasoned that it might be reflected in the burstiness of the spike trains. To this aim, we quantified the local variation (Lv) [38], a measure of the regularity of the temporal spiking dynamics (S2B Fig). This coefficient takes values of 1 if the unit has random spiking dynamics; below 1 if the unit has regular spiking (i.e., oscillatory); and above 1 if the unit is bursty (S2B Fig). In line with our intuition, the Lv very tightly correlated with excitation strength (coefficient estimate = 0.436, 95% CI [0.434, 0.439], p < $10^{-16}$; S2C and S2D Fig, left) and thus also significantly interacted with the IT estimation method effect on REE (coefficient estimate = -0.042, 95% CI [-0.046, -0.039], p < $10^{-16}$; S2D Fig, right). Perhaps more importantly, this analysis revealed that ACF only outperforms iSTTC in regimes with Lv < 1 (crossover point = 1.099), where units are regular-spiking (S2D Fig, right), and thus do not display the exponentially decaying autocorrelation that is a prerequisite to compute an IT.

To evaluate the computational efficiency and scalability of iSTTC, we quantified its per-unit computation time on a modern workstation and assessed how its cost scales with dataset size and signal length. iSTTC incurred a higher per-unit computational cost than ACF (median per unit computation time = 0.2838s for iSTTC and 0.0088s for ACF, coefficient estimate = 0.2715, 95% CI [0.2707, 0.2723], p < $10^{-16}$; S3A right and S3B right Fig). The method difference was strongly moderated by firing rate (coefficient estimate = 0.16, 95% CI [0.1589, 0.1606], p < $10^{-16}$) while excitation strength and intrinsic timescale exerted no significant effect (S3B Fig, right). Nonetheless, the total computation time scaled linearly with the number of units (S3C Fig) and with the signal length (S3D Fig), consistent with the algorithmic complexity of iSTTC. Importantly, even very large synthetic dataset remained computationally tractable: processing $10^5$ unsegmented units with iSTTC required approximately eight hours of compute time on a modern workstation, demonstrating that iSTTC can be applied also to large-scale datasets.

Taken together, this data demonstrates that, on unsegmented spiking data, iSTTC is a better IT estimator than ACF, particularly in regimes that resemble neocortical brain activity, and are characterized by low and bursty firing.

## 2.3 iSTTC outperforms PearsonR in IT estimation accuracy on epoched spiking data

A key advantage of iSTTC over ACF is that the former is insensitive to 0-padding. Consequently, if the data on which IT estimation is based is epoched, as commonly observed in much of the current IT literature [1,6,8,31,39], iSTTC can still

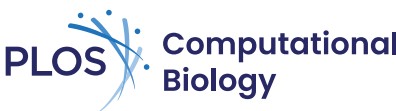

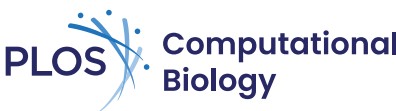

**Fig 3. iSTTC is a better IT estimator than ACF on unsegmented data, particularly for low-firing rate and bursty units.** **(A)** Schematic illustration of the synthetic dataset generation (left), and the underlying parameters with corresponding representative spike train examples (right). **(B)** Definition of the relative estimation error (REE) metric. **(C)** Hexbin plot displaying the difference in REE between iSTTC and ACF method as a function of firing rate and excitation strength (left), and IT and excitation strength (right) (n = $10^5$ single units). Color codes for the median REE difference in each bin, with blue indicating better IT estimation for iSTTC. **(D)** Line plot displaying predicted REE values for iSTTC and ACF as a function of firing rate (left), excitation strength (middle), and IT (right) (n = $10^5$ single units). Shaded areas represent 95% confidence intervals. Y-axes are plotted on a $log_{10}$ scale. In **(C)** and **(D)**, ACF parameters were: bin size = 50 ms, number of lags = 20; iSTTC parameters were: lag shift = 50 ms, dt = 25 ms, number of lags = 20. In **(C)**, asterisks indicate a significant effect of the IT estimation method. In **(D)**, asterisks indicate a significant effect of an interaction between method and firing rate (left), method and excitation strength (middle), and method and IT (right). * $p < 0.05$., *** $p < 0.001$. Generalized linear model with interactions **(C)**, **(D)**.

be readily employed after 0-padding and concatenating the epoched spiking data. Given that ACF cannot be used on epoched data, we compared iSTTC to the PearsonR approach that is commonly used in the literature [1,31,39]. To this aim, we used the same dataset of $10^5$ spike trains that we previously described. We extracted pseudo trials by randomly sampling 40 1s long chunks of spiking data (Fig 4A). The choice of trial number and length was based on values commonly used in the literature [1,8,31,39]. Similarly to our approach in the previous section, we compared the IT estimation accuracy of the two methods by computing the REE for each iteration, and modeled the results with a generalized linear model.

**Fig 4. iSTTC provides better IT estimates from epoched spiking activity. (A)** Schematic illustration of the generation of epoched data based on randomly sampled unsegmented spike trains. Dice icon from svgrepo.com. **(B)** Hexbin plot displaying the difference in REE between iSTTC and PearsonR as a function of firing rate and excitation strength (left), and IT and excitation strength (right) (n = $10^5$ single units, 40 trials x 1000 ms each). Color codes for the median REE difference in each bin, with blue indicating lower error for iSTTC. **(C)** Line plot displaying predicted REE values for iSTTC and PearsonR as a function of firing rate (left), excitation strength (middle), and IT (right) (n = $10^5$ single units, 40 trials x 1000 ms each). Shaded areas represent 95% confidence intervals. Y-axes are plotted on a $log_{10}$ scale. In **(B)** and **(C)**, PearsonR parameters were: bin size = 50 ms, number of lags = 20; iSTTC parameters were: lag shift = 50 ms, dt = 25 ms, number of lags = 20. In **(B)**, asterisks indicate a significant effect of the method. In **(C)**, asterisks indicate a significant effect of an interaction between method and firing rate (left), method and excitation strength (middle), and method and IT (right). * $p < 0.05$., ** $p < 0.01$., *** $p < 0.001$. Generalized linear model with interactions **(B)**, **(C)**.

Across the overwhelming majority of the explored parameter space, iSTTC yielded lower REEs than ACF (coefficient estimate = -0.083, 95% CI [-0.090, -0.077], p < $10^{-16}$; Figs 4B and S4A). On average, iSTTC estimates were 17.5% better than PearsonR (S4A Fig). This difference was not uniform throughout the parameter space (Fig 4B). Accordingly, the effect of IT estimation method significantly interacted with firing rate (coefficient estimate = -0.008, 95% CI [-0.014, -0.001], p = 0.02124; Figs 4C, left and S4A), excitation strength (coefficient estimate = 0.009, 95% CI [0.002, 0.016], p = 0.00985; Figs 4C, middle and S4A) and the value of the IT (coefficient estimate = -0.015, 95% CI [-0.023, -0.009], p < $10^{-6}$; Figs 4C, right and S4A). Thus, estimating ITs with iSTTC results in lower REEs than PearsonR, an advantage that is particularly evident in high firing rate, low excitation strength and high IT values regimes (Figs 4B–4C, S4A, S5A and S5B). Notably, the magnitude of these interaction effects was much more modest than in the unsegmented spiking dataset (compare S2A and S4A). Perhaps most importantly, it is noteworthy that, regardless of the IT estimator that was employed, the REEs obtained from epoched data are roughly one order of magnitude larger than those obtained from unsegmented spiking data.

To evaluate the computational efficiency of iSTTC on epoched data, we followed the same approach as for unsegmented spike trains, quantifying per-unit computation time on a modern workstation and examining how it varied with firing rate, excitation strength, intrinsic timescale, dataset size, and the number of trials. Overall, iSTTC and PearsonR exhibited comparable per-unit computation times (median per unit computation time = 0.0384s for iSTTC and 0.0542s for PearsonR, coefficient estimate = -0.0023, 95% CI [-0.0092, 0.0047], p = 0.52; S6A right and S6B right Fig). Method differences were significantly moderated by firing rate (coefficient estimate = -0.019, 95% CI [-0.026, -0.012], $p < 10^{-7}$) and excitation strength (coefficient estimate = 0.0166, 95% CI [0.0096, 0.0235], p < $10^{-5}$), with iSTTC having faster compute times than PearsonR at higher firing rates, and PearsonR being slightly faster at stronger excitation strength (S6B right Fig). As for unsegmented data, the total computation time scaled linearly with the number of units (S6C Fig) and with the number of trials (S6D Fig), consistent with the algorithmic complexity of iSTTC on trials. Importantly, processing $10^5$ trial-based units with iSTTC required only about two hours, confirming that large-scale datasets remain easily tractable.

Taken together, this data indicates that, after 0-padding and trial concatenation, iSTTC outperforms PearsonR in estimating IT on epoched data. Moreover, the large REEs obtained from epoched data warrant some precaution in interpreting IT estimates obtained with this approach.

## 2.4 Epoched spiking data leads to vastly larger REEs than unsegmented spiking data

Inspired by the large difference on REEs between unsegmented and epoched data, we decided to systematically investigate how the accuracy of IT estimations depends on the amount and type of data that it is based upon. We first compared the REE obtained from the same spiking units in the unsegmented and epoched condition. To create the epoched dataset, we generated pseudo-trials by randomly selecting 40 1s segments of spiking activity for each unit. We confirmed our prior results about epoched data resulting in vastly larger REEs (Fig 5A, left). Irrespective of the IT estimation method, the epoched spiking data yielded a 20% decrease of IT estimates with an REE below 100%, and a 60% decrease of estimates with an REEs below 50% (Fig 5A, middle). In the aggregate, REEs from epoched data were an order of magnitude larger than unsegmented data (Fig 5A, right). Additionally, this analysis confirmed that iSTTC outperforms ACF and PearsonR on unsegmented and epoched data, respectively (Fig 5A). However, the advantage conferred by iSTTC over PearsonR is of limited magnitude when compared to the increase of REE due to epoched spiking data.

To further explore how the amount of data affects the REE of different methods, we systematically explored the effect of varying the amount of unsegmented (from 60s to 10 minutes) data and the number of trials (from 40 to 100 trials) on which we estimated ITs.

Increasing the length of unsegmented spiking data led to a log-linear reduction in the REE of iSTTC and ACF alike (Fig 5B). Across the entirety of the parameter space, iSTTC outperformed ACF by an average of 7% (coefficient estimate = -0.029, 95% CI [-0.031, -0.028], p < $10^{-16}$; S7A Fig) and consistently displayed a higher proportion of IT estimates

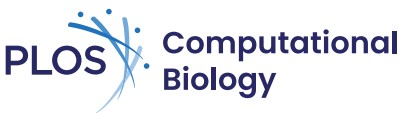

**Fig 5. Relative estimation errors are higher for epoched than unsegmented spiking data. (A)** Comparison of IT estimation accuracy across four methods. Ridgeline plot displaying the distribution of REE. Only IT estimates with REE between 0% and 100% are shown. Percentages indicate the proportion of IT estimates within this interval for each method (left). Scatter plot displaying the percentage of IT estimates with REE falling within progressively narrower intervals (middle). Violin plot displaying the full distribution of REE values for each method. (right). **(B)** Line plot displaying predicted REE values for ACF and iSTTC as a function of signal length (n = $10^5$ single units per signal length). Shaded areas represent 95% confidence intervals. Y-axes are plotted on a $log_{10}$ scale. **(C)** Heatmap displaying the percentage of spike trains with REE within specific intervals for ACF (left) and iSTTC (middle) across varying signal lengths. Color codes for the proportion of spike trains, with warmer colors indicating higher percentages of spike trains. (Right) Heatmap displaying the difference in performance between methods, computed as the difference between ACF and iSTTC. Negative values indicate better performance (lower REE) for iSTTC. Color codes for the magnitude of the difference in percentages. **(D)** Same as **(B)** for PearsonR and iSTTC. **(E)** Same as **(C)** for PearsonR and iSTTC (trials). In **(A)–(D)**, ACF/PearsonR parameters were: bin size = 50 ms, number of lags = 20; iSTTC parameters were: lag shift = 50 ms, dt = 25 ms, number of lags = 20. In **(A)** right, data is presented as median, 25th, 75th percentile, and interquartile range, with the shaded area representing the probability density distribution of the variable. In **(B)**, asterisks indicate a significant effect of interaction between method and signal length. In **(D)**, asterisks indicate a significant effect of an interaction between method and the number of trials. *** $p < 0.001$. Generalized linear model with interactions **(B)**, **(D)**.

within various ranges of the actual IT value (Fig 5C). The effect of IT estimation method interacted with the spike train length, excitation strength and the IT value (signal length: coefficient estimate = -0.006, 95% CI [-0.008, -0.005], p < $10^{-14}$; excitation strength: coefficient estimate = -0.031, 95% CI [-0.032, -0.029], p < $10^{-16}$; IT: coefficient estimate = 0.004, 95% CI [0.002, 0.005], p < $10^{-5}$; S7A Fig). This indicates that iSTTC outperforms ACF particularly in long spiking datasets with low IT values and high excitation strength, and on units with low firing rates.

In line with these results, increasing the number of trials also strongly and log-linearly reduced the REE of iSTTC and PearsonR alike (Figs 5D and S7B). Across the entirety of the parameter space, iSTTC outperformed PearsonR by an average of 14% (coefficient estimate = -0.062, 95% CI [-0.068, -0.057], p < $10^{-16}$; S7B Fig) and consistently displayed a higher proportion of IT estimates within various ranges of the actual IT value (Fig 5E). The effect of IT estimation method interacted with the number of trials and the IT value (number of trials: coefficient estimate = 0.0098, 95% CI [0.005, 0.015], p < $10^{-16}$; IT: coefficient estimate = -0.015, 95% CI [-0.020, -0.01], p < $10^{-16}$; S7B Fig), indicating that the advantage conferred by iSTTC is particularly prominent at low trial number counts and high IT values.

Lastly, iSTTC also converged faster than PearsonR to a stable IT estimate (S7C and S7D Fig). To quantify this, we resampled the same number of trials a varying number of times (50–1000), and assessed the consistency of the estimates. A larger fraction of iSTTC-derived IT values fell within various fixed ranges around the median, indicating greater stability (S7C Fig). In addition, iSTTC showed a lower standard error of the median across all resampling iterations (S7D Fig).

In summary, iSTTC consistently provides more accurate and stable IT estimates than traditional methods across varying data conditions, in particular for datasets with a low number of trials.

### 2.5 Sensitivity of IT estimates to hyperparameter choices

To evaluate whether IT estimation is sensitive to hyperparameter choices, we systematically varied the parameters used to compute ACF/ACF-like across methods (S8A and S8B Fig). To this aim, we used the dataset of $10^5$ unsegmented spike trains from Fig 3 and the trial-based dataset from Fig 4 (40 trials per unit, each 1000 ms long).

For all methods, the autocorrelation function was computed over the interval 0 – 1000 ms. For ACF and PearsonR, which operate on binned data, we varied the bin size to 10, 40, 50, 60, and 100 ms. For iSTTC, we used the same set of values as the lag shift parameter, with corresponding nlags values of 100, 25, 20, 16, and 10, respectively. Hyperparameter choice generally had a small influence on the estimated IT, but the effect was highly heterogeneous across bin sizes (lag shifts). Relative to a bin size (lag shift) of 50 ms, which we used as reference, a 10 ms bin size (lag shift) produced a negative bias in the estimated timescale, whereas 40 ms showed no detectable effect. A 60 ms bin size yielded a small positive shift in unsegmented spike trains, and 100 ms produced a small positive deviation in the trial-based case (S8A and S8B Fig). Importantly, these effects were largely consistent across methods, indicating that the observed variability reflects general properties of the ACF/ACF-like estimation rather than depending on the specific method that is used.

### 2.6 iSTTC improves inclusion rates and fit quality on epoched spiking activity

Current studies on ITs generally impose strict inclusion criteria that limit the analysis to a small percentage of recorded neurons. These criteria commonly include a minimal firing rate of 1 Hz, an absence of empty bins in the averaged binned spiking data, a minimal number of trials in epoched data, a declining ACF/ACF-like in a specific time window, an R-squared value $\geq$ 0.5, and manual inspection of fit quality. [1,8,31–34]. Thus, we set out to investigate whether iSTTC allows the use of a larger proportion of single units for IT estimation, thereby increasing the representativeness and robustness of the measure. We quantified the proportion of rejected units due to a failed or negative R-squared fit, which

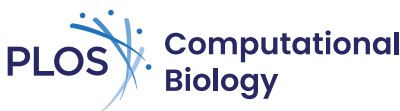

indicates that the model performs worse than predicting the mean of the data. While IT estimations on unsegmented data led to a minimal proportion of rejected units (0.25%), this proportion increased significantly when using the PearsonR method (16.6%) and, to a much less extent, when using iSTTC on epoched data (6.3%) (Fig 6A). The advantage conferred by iSTTC over PearsonR was not uniform throughout the simulation parameters. Rather, the usage of iSTTC reduced the failed fits particularly at low firing rates, high levels of excitation strength and low IT values (Fig 6B).

To further assess whether commonly used quality criteria are valid indicators of estimation accuracy, we took advantage of the REE, which quantifies deviation from the IT ground truth value. We assessed the impact on REE of three quality parameters: the presence of an ACF/ACF-like decline in the 50–200 ms range, the R-squared value, and whether the 95% confidence interval of the IT estimate excludes zero. We found that each parameter was significantly predictive for REE (ACF/ACF-like decline: coefficient estimate=-0.946, 95% CI [-0.954, -0.937], $p < 10^{-16}$; 95% CI excludes 0: coefficient estimate=-1.325, 95% CI [-1.332, -1.318], $p < 10^{-16}$; R-squared: coefficient estimate=-1.613, 95% CI [-1.624, -1.602], $p < 10^{-16}$, Fig 6C), suggesting that they are valid proxies of IT estimation quality. Importantly, since REE cannot be computed on experimental data where the ground truth is unknown, these results support their use to increase the reliability of IT estimates in practice.

Next, we compared how the proportion of units meeting each quality criterion varied across methods. Nearly all units showed an ACF/ACF-like decline in the 50–200 ms range in unsegmented data (88%). This proportion dropped substantially in epoched data: 53.4% of units passed the criterion using PearsonR and 61.4% using iSTTC. Thus, iSTTC retained approximately 8% more units than PearsonR (Figs 6D, left and S9A). A similar pattern was observed for the proportion of units whose IT estimate had a 95% CI that excluded zero. Here, 92% of units met this criterion in unsegmented data, but only by 30.4% with PearsonR and 38% with iSTTC in epoched data, again highlighting an improvement of 7.6% for iSTTC (Figs 6D, right and S9A). Finally, this trend persisted with the R-squared criterion: 96% of units passed in continuous data, compared to 43.9% with PearsonR and 51.6% with iSTTC in epoched data. Once more, iSTTC retained 7.7% more units than PearsonR (Figs 6E and S9A).

To further examine how the amount of data influences the number of included units, we quantified the proportion of units meeting each quality criterion, as well as the proportion of excluded units, across increasing signal lengths and trial counts. In unsegmented data, both ACF and iSTTC showed high pass rates across all quality metrics, but iSTTC consistently resulted in fewer excluded units, particularly at shorter signal length (S10A Fig). In epoched data, the difference between methods became more pronounced. Across all trial counts, iSTTC retained a higher percentage of units compared to PearsonR for each quality criterion and showed a lower percentage of excluded units (S10B Fig). These results highlight that iSTTC is especially advantageous under challenging data conditions, such as short recordings or low trial numbers, where traditional methods are more likely to fail.

Finally, we examined how hyperparameter choice affects inclusion rates. All methods showed an increase in rejected units when using the smallest bin size or lag shift (10 ms) (S10C Fig). On unsegmented data, rejection rates decreased rapidly with increasing bin sizes (lag shifts), with both ACF and iSTTC showing very low exclusion levels (S10C left Fig). On epoched data, rejection rates also declined as bin size increased, however, PearsonR consistently exhibited higher exclusion rates than iSTTC across all bin sizes (S10C right Fig).

Taken together, these findings indicate that iSTTC allows a larger proportion of units to meet commonly used quality criteria, thereby increasing the yield and representativeness of IT estimates.

## 2.7 iSTTC and aABC are designed for different use cases

To directly compare iSTTC with adaptive Approximate Bayesian Computation (aABC), a recently introduced Bayesian-based framework to estimate ITs [29], we focused on a subset of 10,000 units from the synthetic dataset used in Fig 5. aABC is a method that improves IT estimation by optimizing the step of extracting timescales based on the autocorrelation function (step 2 of IT estimation) by leveraging generative models to simulate synthetic signals across a range of candidate timescales, compute their autocorrelation functions, and return a posterior distribution over timescales. In contrast,

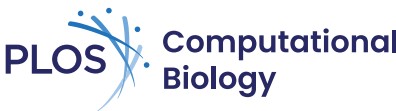

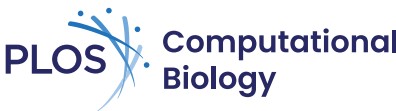

**Fig 6. iSTTC allows for the inclusion of more units than PearsonR. (A)** Bar plot displaying the percentage of excluded units across four methods. Color codes for the exclusion reason, with dark grey indicating failed exponential fits and light grey indicating negative R-squared values of the exponential fit (n=23348 excluded fits across four methods, n=248 excluded fits ACF, n=238 excluded fits iSTTC, n=16594 excluded fits PearsonR, n=6268 excluded fits iSTTC (trials)). **(B)** Kernel density plot displaying the distribution of excluded units as a function of firing rate (left), excitation strength (middle), and IT (right) (n=4469 units both methods failed, n=12125 units only PearsonR, n=1799 units only iSTTC). **(C)** Violin plot displaying REE values

for exponential fits where the autocorrelation function declined (vs. not declined) in the 50–200 ms range (left), and for fits where the 95% confidence interval of the estimated IT included vs. excluded zero (middle) (n = 376652 fits across four methods). Line plot displaying predicted REE values as a function of R-squared (right) (n = 376652 fits across four methods). Shaded areas represent 95% confidence intervals. Y-axes are plotted on a $log_{10}$ scale. **(D)** Bar plots displaying the percentage of units with autocorrelation function decline in the 50–200 ms range (left, n = 291395 fits across four methods), and the percentage of units with 95% confidence intervals excluding zero (right, n = 253595 fits across four methods), across four methods. **(E)** Kernel density plot displaying the distribution of R-squared values (left), and bar plot displaying the percentage of units with R-squared $\geq$ 0.5 across four methods (right, n = 287967 fits across four methods). In **(C)**, left and middle, data are presented as median, 25th, 75th percentile, and interquartile range, with the shaded area representing the probability density distribution of the variable. In **(C)**, asterisks indicate a significant effect of the factor on REE. *** $p < 0.001$. Generalized linear models **(C)**.

iSTTC targets step 1 of the IT estimation pipeline, and provides a more robust and less biased estimate of the ACF-like, as outlined in the Introduction. Thus, the two methods act on different components of the IT estimation process and, in principle, could be combined, although this lies beyond the scope of the present work.

We first compared IT estimation accuracy across methods on unsegmented and epoched data in this reduced dataset (S11A Fig). Working on a reduced dataset was necessary because of the long compute time of aABC (see below). Of note, although it is in principle possible to apply aABC to long, continuous spike trains, its computational cost renders such applications impractical. In addition, in long time series the bias in autocorrelation estimates is negligible [40]. As a summary statistic in aABC, we used the "comp_cc" provided in the abcTau toolbox, which computes the autocorrelation function in the time domain from binned spike trains. On this data, aABC yielded a lower fraction of units with REE < 100% and a higher median REE than full-signal iSTTC and ACF, indicating that optimizing the timescale extraction step alone does not compensate for inaccuracies arising from the ACF estimation (S11A Fig). In contrast, on epoched data, aABC outperformed both iSTTC and PearsonR. This improvement primarily reflected a suppression of the heavy right tail of the REE distribution: aABC produced fewer extreme overestimates than the other trial-based approaches (S11A Fig).

To better characterize the relative strengths of aABC and iSTTC on epoched data, we directly compared their performance as a function of the ground-truth IT (S11B Fig). In regimes with low IT values, iSTTC produced smaller REEs than aABC, suggesting that a more accurate ACF-like estimate is particularly beneficial when intrinsic timescales are short. For intermediate and long IT values, aABC tended to outperform iSTTC (S11B Fig).

We next assessed the impact of each method on unit inclusion. While ACF and iSTTC yielded virtually no rejected units, aABC produced a non-negligible fraction of units (6.5%) for which the algorithm did not converge to an acceptable fit (S11C Fig). This additional rejection step has no analogue when using iSTTC or ACF alone and effectively reduces the usable sample size. Of note, the rejection rate in aABC depends on the selected summary statistic, and employing iSTTC as a summary statistic within the aABC framework could result in comparable rejection rates.

Finally, we compared computational costs (S11D Fig). The median runtime per unit was 212s for aABC, compared to 0.04s for iSTTC, a > 5,000-fold difference. In practical terms, running aABC on as few as 500 units requires on the order of 30 hours on a modern workstation, whereas iSTTC completes in under a minute. This scaling makes aABC impractical for large-scale SUA datasets, and explains why recent applications of aABC have focused on population-level signals, for example by aggregating spikes across neurons into multi-unit activity before estimating timescales [6,29]. Of note, the runtime of aABC depends in part on the chosen summary statistic. Replacing the time-domain "comp_cc" implementation with the FFT-based autocorrelation reduces the cost of the summary-statistics step by approximately 13.6-fold (92.6%). However, even under the conservative assumption that this step dominates runtime, aABC would remain orders of magnitude slower than direct exponential fitting approaches.

Taken together, these results show that iSTTC and aABC address complementary steps of the IT estimation pipeline and are designed for different use cases. On unsegmented data, iSTTC (and ACF) applied directly to SUA remains markedly more accurate and inclusive than any trial-based approach, including aABC. On epoched data, aABC outperforms iSTTC and PearsonR primarily by avoiding very large errors. However, its computational cost and rejection rates under

our chosen parameter configuration make it best suited for population-level applications rather than single-unit analysis. Conceptually, the two approaches are not in conflict: iSTTC improves the robustness of the ACF-like estimate (step 1), whereas aABC refines the timescale extraction stage (step 2). In principle, combining iSTTC-derived ACF-like functions with aABC-based timescale extraction could further improve IT estimation by leveraging the complementary strengths of both approaches.

## 2.8 iSTTC provides more stable, robust and inclusive IT estimates on experimental data

To assess the performance of iSTTC on unsegmented and epoched experimental data, we analyzed a subset of the Visual Coding Neuropixels dataset [2,41]. The dataset comprised 5,775 single units from 26 mice (Figs 7A and S12A). Recordings were obtained from six cortical areas: primary visual cortex (V1), lateromedial area (LM), anterolateral area (AL), rostrolateral area (RL), anteromedial area (AM), and posteromedial area (PM), and two thalamic areas: the lateral geniculate nucleus (LGN) and the lateral posterior nucleus (LP). To estimate baseline ITs, we used data from the Functional Connectivity sessions, which included 30 minutes of spontaneous activity during gray screen presentation. We then artificially created trials by randomly selecting 40 1s-long segments from each 30-minute spike train, following the same approach used in the synthetic dataset. Firing rates ranged from 0.06 to 69.7 Hz (median: 3.9 Hz), and local variance from 0.14 to 2.16 (median: 0.86) (Figs 7A and S12A).

We estimated ITs at the brain area level using two unsegmented-signal methods and two trial-based methods. Full-signal methods yielded consistent estimates across areas with relatively narrow confidence intervals and a well-preserved hierarchical ordering of ITs, consistent with previous results [36] (S12B–S12D Fig). Conversely, trial-based methods produced more variable estimates (S12B Fig) and hierarchical ordering of ITs (S12C–S12D Fig).

To further assess the stability of trial-based estimates, we repeated the trial sampling procedure 50 times per unit and recomputed area-level ITs. Resampling led to high variability in the estimates across trial sets (Fig 7B; the black dots mark the values from the original sampling iteration used in S12B–S12D Fig). Together, these results highlight how the instability of trial-based methods can lead to inconsistent results and hierarchical ordering of ITs.

Next, we compared IT estimates at the single-unit level. ACF and iSTTC produced similar results with no significant differences between them (coefficient estimate = 0.030, 95% CI [-0.023, 0.085], p = 0.264). For this reason, and in light of the much greater accuracy of IT estimates on synthetic unsegmented data, we used these values as pseudo ground-truth ITs to compare with trail-based IT estimates. First, we noticed that IT estimates from both trial-based methods differed significantly from ACF (PearsonR: coefficient estimate = 0.655, 95% CI [0.601, 0.709], $p < 10^{-16}$; iSTTC trials: coefficient estimate = 0.624, 95% CI [0.570, 0.678], $p < 10^{-16}$; Figs 7C left and S12E). Importantly, when modeling the pseudo-REEs we found that, on average, iSTTC yielded pseudo-REEs that were 15% smaller than PearsonR (coefficient estimate = -0.071, 95% CI [-0.117, -0.024], p = 0.0028; Figs 7C middle and S12F). Along the same lines, iSTTC led to a larger (5%) proportion of IT estimates with a pseudo-REE that was within a range of 100% to 25% of the IT estimated by iSTTC (Fig 7C, right).

We next quantified the effect of signal duration and trial number on IT estimated from unsegmented and epoched data, respectively. To this aim, we systematically varied the signal length (from 1 to 20 minutes) and the trial count (from 40 to 100 trials), and computed the pseudo-REE with respect to the IT estimated using iSTTC on the full-signal. Similarly to synthetic data, increasing signal length led to a log-linear decrease in pseudo-REE. Importantly, iSTTC outperformed ACF across the entirety of the dataset (7% lower REEs, coefficient estimate = -0.031, 95% CI [-0.047, -0.016], $p < 10^{-5}$; S13A Fig), an advantage that was particularly evident on short signals (Figs 7D and S13A). Further, iSTTC consistently estimated a larger proportion of IT with a pseudo-REE that fell within various ranges of the IT estimated on the full signal (Fig 7E).

On epoched data, increasing the number of trials also led to a log-linear decrease in pseudo-REE, but no differences between iSTTC and PearsonR reached statistical significance (coefficient estimate = -0.025, 95% CI [-0.054, 0.004],

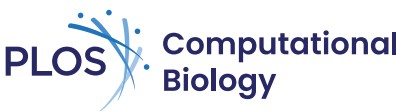

**Fig 7. iSTTC provides more stable, robust and inclusive IT estimates than ACF and PearsonR also on experimental data. (A)** Schematic representation of Neuropixels recordings from the six visual cortical areas (V1, LM, RL, AL, PM, AM) and the two thalamic areas (LGN and LP), image source

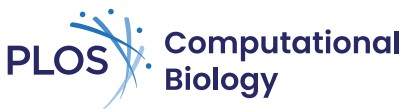

[2] (left), violin plots displaying the single units firing rate (top right) and local variation (bottom right). Mouse icon from scidraw.io (DOI https://zenodo.org/records/3925991). **(B)** Scatter plots displaying ITs at the brain area level. Black dots indicate area-level ITs used for the analyses in S12B–S12D Fig. Grey dots represent individual brain-area IT estimates for the trial-based methods across different sampling iterations (n = 50 samples). **(C)** Violin plots displaying the estimated ITs (n = 3053 single units, only units for which all methods produced an IT estimate are included) as a function of the estimation method (left). Violin plots displaying pseudo-REE as a function of the estimation method (middle). Scatter plot displaying the percentage of spike trains with pseudo-REE falling within progressively narrower bounds (right). **(D)** Line plot displaying predicted pseudo-REE values for iSTTC and ACF as a function of signal length (n = 5674 single units per signal length, only units for which both methods produced an IT estimate for all signal lengths are included). Shaded areas represent 95% confidence intervals. Y-axes are plotted on a $log_{10}$ scale. **(E)** Heatmap displaying the percentage of spike trains with pseudo-REE within specific intervals for ACF (left) and iSTTC (middle) across varying signal lengths. Color codes for the proportion of spike trains, with warmer colors indicating higher percentages of spike trains. (Right) Heatmap displaying the difference between ACF and iSTTC. Negative values indicate better performance (lower pseudo-REE) for iSTTC. Color codes for the magnitude of the difference. **(F)** Same as **(D)** for PearsonR and iSTTC (n = 4588 single units per number of trials, only units for which both methods produced an IT estimate for all numbers of trials are included). **(G)** Same as **(E)** for PearsonR and iSTTC. In **(B)**–**(G)**, ACF/PearsonR parameters were: bin size = 50 ms, number of lags = 20; iSTTC parameters were: lag shift = 50 ms, dt = 25 ms, number of lags = 20. In **(A)** and **(C)** left and middle, data is presented as median, 25th, 75th percentile, and interquartile range, with the shaded area representing the probability density distribution of the variable.In **(C)**, asterisks indicate a significant effect of the method. In **(C)** left, ACF is used as a reference. In **(C)** middle, PearsonR is used as a reference. In **(D)**, asterisks indicate a significant effect of signal length. ** $p$ < 0.01, *** $p$ < 0.001. Generalized linear models with interactions **(C)**, **(D)**, and **(F)**.

p = 0.0894; Figs 7F and S13B). However, pseudo-REEs of ITs estimated with iSTTC consistently fell at a higher proportion within various ranges of the IT estimated on the full signal (Fig 7G). Moreover, iSTTC converged faster to an accurate IT estimation. To quantify this, we resampled the same dataset 50–1000 times with different and randomly selected trials, and assessed the consistency of the estimates. A larger fraction of iSTTC-derived IT values fell within various ranges around the median, indicating greater stability (S13C Fig). In addition, iSTTC showed a lower standard error of the median across resampling iterations (S13D Fig).

Lastly, we evaluated whether iSTTC allows to include a larger proportion of units also on experimental data. To this aim, we quantified the proportion of rejected units and the proportion of units meeting the same three quality criteria that we used for the synthetic data. In contrast to the synthetic dataset, we observed a high proportion of excluded units also in unsegmented data, with minimal differences between ACF and iSTTC (21–22%). Similarly to our previous results, this proportion was substantially higher on epoched data. Also in this case, iSTTC outperformed PearsonR, yielding a rejection rate of 28.3% compared to 40.6%, a decrease of 12.4% (S14A Fig). When comparing PearsonR and iSTTC on epoched data, iSTTC allowed to include a larger proportion of units with low firing rates and bursty firing patterns (S14B Fig). In line with these results, on epoched data, iSTTC also retained a higher percentage of units passing each quality criterion, including ACF/ACF-like decline in 50–200 ms range (5.3% increase for iSTTC), 95% CI excludes 0 (3.6% increase for iSTTC), and R-squared $\geq$ 0.5 (4.7% increase for iSTTC) (S14C–S14D Fig). On unsegmented data, iSTTC and ACF performed similarly (S14C–S14D Fig). These findings show that, also on epoched experimental data, iSTTC enables the use of a larger proportion of recorded units.

In conclusion, also on experimental data, iSTTC provides more stable, robust and inclusive IT estimates than ACF and PearsonR. Moreover, IT estimates based on unsegmented data are more robust and consistent than trial-based methods.

## 3 Discussion

In this study we introduce iSTTC, a novel method for accurately estimating ITs from SUA recordings. We demonstrate the advantages of iSTTC over currently used methods using synthetic and experimental data. Under a wide variety of conditions, iSTTC provides more accurate and stable IT estimates. Further, iSTTC accommodates both unsegmented and epoched data without bias, and generally allows for the inclusion of a larger number of single units. Together, these properties significantly enhance the accuracy, representativeness and robustness of IT estimations in neural circuits. A fundamental limitation of IT research, including our work, is that ground-truth ITs are only available in synthetic data. For experimental data, we cannot directly verify absolute IT values and therefore interpret the benefits of iSTTC mainly in terms of robustness, stability and inclusivity of single units.

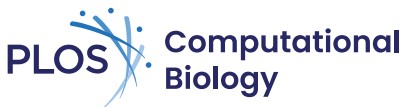

iSTTC is an extension of STTC [35], adapted to address the specific challenges of IT estimation. Unlike traditional ACF approaches, iSTTC directly quantifies the autocorrelation-like function on binary spiking data without binning or smoothing. This reduces the number of arbitrary parameters required for the analysis. Crucially, iSTTC is also insensitive to 0-padding, and can therefore be seamlessly applied also to epoched data. When tested on synthetic data, iSTTC recovers ITs more accurately than other state-of-the-art methods, both on unsegmented and epoched signals. On unsegmented data, these advantages are especially pronounced under biologically realistic conditions, such as sparse firing and strong recurrent connectivity. Lastly, on epoched data, iSTTC produces a larger amount of ACF-like estimates that pass commonly used inclusion criteria, significantly expanding the proportion of neurons suitable for analysis. This increased single-unit inclusivity enhances statistical robustness, ensures IT estimates that better represent the underlying neural circuit, and allows for the study of individual neuron ITs on a larger population of units. When investigating a state-of-the-art experimental dataset, iSTTC required shorter recording durations than traditional ACF methods to converge to stable solutions. On epoched experimental data, iSTTC remained more robust, converging more rapidly to estimates closer to those derived from unsegmented recordings. Further, we confirmed that substantially more neurons met inclusion criteria under iSTTC then PearsonR, thus producing more representative brain-area IT estimates.

An additional key insight from our analysis is the inherent instability in ITs estimated from epoched data, a common practice in the field. Our analyses on simulated and experimental data demonstrate that short trial lengths or limited trial numbers drastically reduce the reliability of IT measurements. Overall, estimating ITs on epoched data with a number and duration of trials that is similar to those employed in the literature [1,8,31,39] leads to a roughly ten-fold increase in REEs when compared to IT estimated on unsegmented data. This warrants caution in interpreting ITs computed with this approach. While iSTTC mitigates this instability and improves estimation accuracy, we emphasize the importance of using either long uninterrupted recordings or, if epoched data is unavoidable, ensuring maximal trial length and number.

The methodological advances introduced by iSTTC have important implications for future research on ITs. Reliable estimation of ITs is critical for understanding how they are regulated and to link them to cognitive and behavioral outcomes. Current theoretical frameworks suggest IT variability across brain areas underpins functional specialization [4]. By enabling more accurate and inclusive measurement of ITs, iSTTC provides a powerful tool to empirically test these theories across diverse neural systems. Moreover, given the wide heterogeneity of single-neuron ITs also within brain areas [8,16,21,39,40], and the emerging evidence linking IT modulation to single neuron properties [4,39], we anticipate that a robust metric which increases the proportion of neurons passing inclusion criteria like iSTTC will be particularly insightful to deepen our understanding of the principles governing ITs. At the same time, also iSTTC still fails to estimate ITs from a small subset of single units. Future experimental work should determine whether these excluded units correspond to specific biological subpopulations whose systematic omission might bias population-level conclusions.

In conclusion, iSTTC expands the toolbox for IT estimation, overcomes several shortcomings of the current methods, and enhances both accuracy and representativeness of IT estimations. These improvements might lead to deeper insights into the mechanisms regulating ITs, ultimately bringing us closer to one of the fundamental goals of neuroscience: mapping behavioral actions onto neural circuits.

## 4 Materials and methods

### 4.1 Datasets

**4.1.1 Synthetic dataset.** To generate synthetic spike trains with a specific intrinsic time constant, we implemented a univariate Hawkes point process with an exponential kernel, using Ogata's thinning algorithm [42,43].

Hawkes process is defined to be a self-exciting temporal point process whose conditional intensity function $\lambda(t)$ is defined as:

$$\lambda(t) \; = \; \mu \; + \; \sum_{t_i < t} g(t - t_i) \, ,$$

(3)

where $\mu > 0$ is the baseline intensity and $g(\Delta)$, where $\Delta = t - t_i$, is an exponential excitation kernel defined as:

$$g(\Delta) \;=\; \frac{\alpha}{\tau_{\text{kernel}}} \, \exp\left(-\frac{\Delta}{\tau_{\text{kernel}}}\right), \quad \Delta \geq 0, \tag{4}$$

with $\alpha \in [0, 1)$ governing the total integrated excitation (ensuring stability) and $\tau_{\text{kernel}}$ as the kernel's time constant. Hence, explicitly:

$$\lambda(t) \;=\; \mu \;+\; \sum_{t_i < t} \frac{\alpha}{\tau_{\text{kernel}}} \, \exp\left(-\frac{t - t_i}{\tau_{\text{kernel}}}\right). \tag{5}$$

Because $\alpha < 1$, the process remains subcritical and its long–term rate converges to $\frac{\mu}{1-\alpha}$.

For a Hawkes process with an exponential kernel, the resulting spike train has an autocorrelation function that decays exponentially with the lag $\Delta$. More precisely, the theoretical ACF of the process takes the form

$$\text{ACF}(\Delta) \;\propto\; \exp\left(-\frac{|\Delta|}{\tau_{\text{ACF}}}\right), \tag{6}$$

with an autocorrelation time constant given by the closed-form relationship [44,45]:

$$\tau_{\text{ACF}} \;=\; \frac{\tau_{\text{kernel}}}{1 - \alpha}. \tag{7}$$

Consequently, by choosing

$$\tau_{\text{kernel}} \;=\; \tau(1 - \alpha), \tag{8}$$

our simulations yield spike trains whose ground-truth intrinsic timescale is $\tau_{\text{ACF}} = \tau$. This value $\tau$ is the target time constant that the different estimation methods aim to recover.

For each spike train simulation, we specified four parameters: a target stationary firing rate $f$ (in Hz), a desired time constant $\tau$ (in ms), an excitation strength $\alpha$ (dimensionless, with $\alpha < 1$ for process stability), and a total simulation duration $d$ (in ms). The kernel time constant was computed as $\tau_{\text{kernel}} = \tau(1 - \alpha)$, and the baseline intensity $\mu$ in units of Hz was set to $\mu = f(1 - \alpha)$.

The method is implemented in the *simulate_hawkes_thinning* function (available at https://github.com/iinnpp/isttc).

The generated datasets are described in the Table 3.

**4.1.2 Mouse dataset.** We analyzed a subset of the Visual Coding Neuropixels dataset, which is publicly available via the Allen Brain Observatory [2,41]. This dataset was previously used to calculate intrinsic timescales [36]. It comprises extracellular electrophysiological recordings from mouse brain acquired with Neuropixels probes. During the experiments, head-fixed mice were presented with a variety of visual stimuli. The dataset includes two sets from the two experimental pipelines, Functional Connectivity and Brain Observatory 1.1, which differ in their stimulus sequences. To compute baseline intrinsic timescales, we used data from the Functional Connectivity set, which contains a block of around 30 minutes of spontaneous activity recorded while the animals viewed a gray screen. For the analysis, we trimmed all recordings to 30 minutes. Recordings span six cortical areas (primary visual cortex (V1), lateromedial (LM), anterolateral (AL), rostrolateral (RL), anteromedial (AM), and posteromedial (PM)) and two thalamic areas (lateral geniculate nucleus (LGN) and lateral posterior nucleus (LP)).

For analysis, we only included single units satisfying the following quality criteria (in agreement with [36]:

• presence ratio ≥ 0.9 (presence ratio 0.9 or higher means that the unit was present at least 90% of the recorded time),

**Table 3. Summary of synthetic datasets.**

| Dataset | Parameters | Dataset file | Related figures |
|---|---|---|---|
| Parametric dataset | $f_{min}$ = 0.01 Hz, $f_{max}$ = 10 Hz, $alpha_{min}$ = 0.1, $alpha_{max}$ = 0.9, $tau_{min}$ = 50 ms, $tau_{max}$ = 300 ms, d = 600 seconds, number of spike trains = 100000. | spike_trains.pkl | Fig 3, Fig 4, Fig 5A, Fig 6 |
| | Trial generation: number of trials = 40, trial duration = 1000 ms. | | |
| | A total of 100 000 independent parameter sets were generated by sampling $f$, $tau$, $alpha$ uniformly within the predefined intervals. | | |
| Parametric datasets with a varying number of trials | For each spike train in the previously generated dataset (600 seconds in duration), four trial sets were created, each consisting of a different number of 1000 ms trials (40, 60, 80, and 100 trials). | trials40.pkl, trials60.pkl, trials80.pkl, trials100.pkl | Fig 5B–5C |
| Parametric datasets with a varying signal length | For each spike train in the previously generated dataset (600 seconds in duration), four truncated versions of lengths 60 seconds, 150 seconds, 300 seconds, and 450 seconds were generated. | length60.pkl, length150.pkl, length300.pkl, length450.pkl | Fig 5D–5E |

- ISI (interspike interval) violations ≤ 0.5 (ISI violations 0.5 of lower means that contaminating spikes are occurring at most at roughly half the rate of "true" spikes for the unit),

- amplitude cutoff ≤ 0.01 (amplitude cutoff of 0.01 or lower indicates that at most 1% of spikes are missing from the unit).

After applying these criteria, 5775 units remained, with a minimum of n = 254 units (LGN) and n = 12 mice (LGN) and a maximum of n = 1063 units (V1) and n = 24 mice (V1 and AM) per brain area.

## 4.2 Autocorrelation function estimation

We used four methods to compute the autocorrelation function. We reserve the term "ACF" for the classic spike-count autocorrelation computed on binned spike trains, and use "ACF-like" to refer to surrogate functions that play the same role in the IT estimation pipeline (e.g. iSTTC- and PearsonR-derived curves). Two methods were applied to the full spike train, the "classic" autocorrelation (ACF) on binned spike trains and iSTTC on non-binned spike trains. The other two were trial-based, using PearsonR on binned spike trains and iSTTC on concatenated non-binned spike trains.

**4.2.1 "Classic" autocorrelation - ACF.** To compute the ACF on binned spiking data, we used the following equation:

$$r(k) = \frac{\sum_{t=k+1}^{N}(x_t - \bar{x})(x_{t-k} - \bar{x})}{\sum_{t=1}^{N}(x_t - \bar{x})^2},$$

(9)

where $r(k)$ is the autocorrelation at $k$ lag and $N$ is the total number of time points (bins) in the signal. We used the `stats-models.tsa.stattools.acf` function for ACF calculation.

**4.2.2 i(ntrinsic)STTC.** We adapted STTC (Spike time tiling coefficient) [35] to compute an ACF-like of spike trains without binning.

iSTTC requires three parameters: the number of lags $nlags$, which specifies the total number of time lags over which the ACF is estimated; $lag\_shift$, which defines the increment between successive lags; and $\Delta t$, within which the iSTTC evaluates spike train pair correlations. The method is implemented in the acf_sttc function (available at https://github.com/iinnpp/isttc).

## Algorithm 1. iSTTC-based autocorrelation function.

**Require:** Spike times $S$, signal length $T$, number of lags $nlags$, lag increment $lag\_shift$, window size $\Delta t$
**Ensure:** Autocorrelation values $\text{ACF}[0...nlags]$
  1: Construct time lag list: $lag\_shift\_list \leftarrow \{lag\_shift, 2lag\_shift, ..., nlags * lag\_shift\}$
  2: Compute zero-lag autocorrelation:
$$\text{ACF}[0] \leftarrow \text{STTC}(S, S, \Delta t, T)$$
  3: **for** each $lag$ in lag_shift_list **do**
  4:   Select spikes occurring before $T - lag$:
$$A \leftarrow \{ t \in S \mid t < T - lag \}$$
  5:   Select spikes occurring after the $lag$:
$$B \leftarrow \{ t \in S \mid t \geq lag \}$$
  6:   Realign spikes in B:
$$B^{\text{realigned}} \leftarrow \{ t - lag \mid t \in B \}$$
  7:   Compute iSTTC:
$$\text{ACF}[lag] \leftarrow \text{STTC}(A, B^{\text{realigned}}, \Delta t, T - lag)$$
  8: **end for**
  9: **return** ACF

Let $N$ denote the total number of spikes in $S$. For each time lag, the algorithm (i) selects the spikes falling within the time window, (ii) realigns the spike train by subtracting the time lag, and (iii) computes the STTC between the resulting spike trains. Each of these steps runs in time proportional to the number of spikes, and the STTC implementation itself is linear in the spike counts of the two input trains. Thus, the per-lag cost is $\mathcal{O}(N)$, and evaluating all $nlags$ lags yields a total time complexity of $\mathcal{O}(nlagsN)$ for one neuron. In our analyses, $nlags$ is fixed by the timescale range of interest and does not increase with recording duration, so runtime effectively scales linearly with the number of spikes (and therefore with firing rate and recording duration).

### 4.2.3 PearsonR trial average.

We implemented the trial-averaged ACF-like function of spike counts (binned spike trains) using Pearson's correlation coefficient as previously described [1,2,8,31,32].

The PearsonR trial average method requires as a parameter the number of lags $n_{\text{lags}}$, which specifies the total number of time lags over which the ACF is estimated. An additional parameter is the *bin size*, which is used to bin the spike trains before applying PearsonR. The method is implemented in the *acf_pearsonr_trial_avg* function (available at https://github.com/iinnpp/isttc).

Given a set of binned spiking data from multiple trials, represented as:

$$X = \{X_1, X_2, ..., X_T\}, \quad X_t \in \mathbb{R}^N, \tag{10}$$

where $X_t$ is the binned spikes time series data for trial $t$, and $N$ is the number of time bins, the ACF for *t*he first $n_{\text{lags}}$ time lags was computed as follows:

**Step 1: Extract relevant time series.** We define a subset of the time series that includes the first $n_{\text{lags}}$ time bins across trials:

$$X' = X[:, 0 : n_{\text{lags}}], \tag{11}$$

where $X'$ is a matrix of size $T \times n_{\text{lags}}$, with $T$ trials and $n_{\text{lags}}$ time bins.
**Step 2: Compute Pearson correlation for each lag.** For each pair of time points $i$ and $j$, where $j > i$, the Pearson correlation coefficient is computed as:

$$r_{i,j} = \frac{\sum_{t=1}^{T}(X'_{t,i} - \bar{X}'_i)(X'_{t,j} - \bar{X}'_j)}{\sqrt{\sum_{t=1}^{T}(X'_{t,i} - \bar{X}'_i)^2} \cdot \sqrt{\sum_{t=1}^{T}(X'_{t,j} - \bar{X}'_j)^2}}, \tag{12}$$

where $r_{i,j}$ represents the Pearson correlation coefficient between time bins $i$ and $j$, $X'_{t,i}$ is the value of the time series at bin $i$ for trial $t$, $X'_{t,j}$ is the value of the time series at bin $j$ for trial $t$, $\bar{X}'_i$ is the mean across trials for bin $i$, $\bar{X}'_j$ is the mean across trials for bin $j$.

These values are stored in an ACF matrix $R$ of size $n_{\text{lags}} \times n_{\text{lags}}$, with:

$$R[i, j] = r_{i,j}, \quad \text{for } j > l, \tag{13}$$

where the diagonal elements are set to $R[i, i] = 1$ (self-correlation at time lag 0).

**Step 3: Compute the trial-averaged ACF-like function.** To obtain the trial-averaged ACF-like function, we compute the mean correlation for each lag $k$, considering the values along the diagonals of the ACF-like matrix:

$$ACF(k) = \frac{1}{n_k} \sum_{i=1}^{n_k} R[i, i + k], \tag{14}$$

where $n_k$ is the number of valid correlations available for each lag $k$.

The result is a one-dimensional autocorrelation function of length $n_{\text{lags}}$.

**4.2.4 iSTTC (trials).** In the iSTTC for epoched signals, all trials for a given unit are concatenated, with zero padding inserted between trials, to form an unsegmented spike-train signal. This concatenated signal is then processed analogously to iSTTC.

The iSTTC for trials requires four parameters: the number of lags $n_{\text{lags}}$, which specifies the total number of time lags over which the ACF-like is estimated; *lag_shift*, which defines the increment between successive lags; $\Delta t$, within which the STTC evaluates spike train pair correlations; and *zero_padding_len*, which is the length of zero padding appended to each concatenated trial. The method is implemented in the acf_sttc_trial_concat function (available at https://github.com/iinnpp/isttc). In addition to this function, we use a modified STTC implementation (sttc_fixed_2t) that allows the term $T$ to be supplied explicitly rather than computed from the concatenated spike train. This prevents zero padding from artificially altering $T$ and ensures that the autocorrelation reflects only the spike timing structure.

**Algorithm 2. iSTTC-based autocorrelation across concatenated trials.**

**Require:** Spike trains $\{S^{(m)}\}_{m=1}^{N}$ (N number of trials), number of lags *nlags*, lag increment *lag_shift*, window size $\Delta t$, trial duration $T_{\text{trial}}$, padding duration $T_{\text{pad}}$
**Ensure:** Autocorrelation values $ACF[0 \dots nlags]$
 1: Concatenate all trials with padding:
$$\tilde{S}_0 \leftarrow ConcatenateWithPadding(\{S^{(m)}\}_{m=1}^{N}, T_{\text{pad}})$$
 2: Compute T for lag *0*:
$$T_0 \leftarrow AverageT(\{S^{(m)}\}_{m=1}^{N}, T_{\text{trial}}, \Delta t$$
 3: Compute zero-lag autocorrelation:
$$ACF[0] \leftarrow STTCFixed2T(\tilde{S}_0, \tilde{S}_0, \Delta t, T_0, T_0)$$
 4: Construct time lag list:
$$lag\_shift\_list \leftarrow \{lag\_shift, 2lag\_shift, \dots, nlags * lag\_shift\}$$
 5: **for** each *lag* in lag_shift_list **do**
 6:   Initialize shifted trial spike trains: $\{S_{1,lag}^{(m)}\}$ and $\{S_{2,lag}^{(m)}\}$
 7:   **for** $m = 1$ to $N$ **do**
 8:     $S_{1,lag}^{(m)} \leftarrow \{t - lag \mid t \in S^{(m)}, t \geq lag\}$ Shifted and realigned segment
 9:     $S_{2,lag}^{(m)} \leftarrow \{t \in S^{(m)} \mid t < T_{\text{trial}} - lag\}$ Unshifted segment
10:   **end for**
11:   Concatenate shifted trials with padding:
$$\tilde{S}_{1,lag} \leftarrow ConcatenateWithPadding(\{S^{(m)}_{1,lag}\}, T_{\text{pad}}$$
$$\tilde{S}_{2,lag} \leftarrow ConcatenateWithPadding(\{S^{(m)}_{2,lag}\}, T_{\text{pad}}$$
12: Compute T:

$$T_{1,lag} \leftarrow AverageT(\{S^{(m)}{}_{1,lag}\}, T_{\mathrm{trial}}-lag, \Delta t$$
$$T_{2,lag} \leftarrow AverageT(\{S^{(m)}{}_{2,lag}\}, T_{\mathrm{trial}}-lag, \Delta t$$

13: Compute iSTTC:

$$\mathrm{ACF}[lag] \leftarrow \mathtt{STTCFixed2T}(\tilde{S}_{1,lag}, \tilde{S}_{2,lag}, \Delta t, T_{1,lag}, T_{2,lag})$$

14: **end For**
15: **return** ACF

Let $N$ denote the total number of spikes across all trials. For each time lag, acf_sttc_trial_concat (i) constructs shifted and unshifted spike segments for all trials, (ii) concatenates these segments with padding, (iii) recomputes the $T_{1,lag}$ and $T_{2,lag}$, and (iv) evaluates the STTC with fixed $T$ on the concatenated spike trains. Each of these steps runs in time proportional to the number of spikes, and the STTC implementation itself is linear in the spike counts of the two input trains. Thus, the per-lag cost is $\mathcal{O}(N)$, and evaluating all *nlags* lags yields a total time complexity of $\mathcal{O}(nlagsN)$ for one neuron across all trials. As in the unsegmented spike train case, *nlags* is fixed by the timescale range of interest, so runtime scales approximately linearly with the total number of spikes across trials and with the number of trials.

## 4.3 Intrinsic timescale estimation

To estimate the intrinsic time constant $\tau$ from the signal autocorrelation function, we fit the function to a single-exponential function defined as:

$$y(t) = a(e^{-bt} + c), \tag{15}$$

where $a$ is the amplitude, $b$ is the decay rate, and $c$ is an offset parameter. The time constant $\tau$, which characterizes the decay rate, is given by:

$$\tau = \frac{1}{b}, \tag{16}$$

We estimate the parameters $a, b, c$ using non-linear least squares fitting with the curve_fit function from SciPy. To ensure numerical stability, we constrain $b$ to be strictly positive during fitting: $b > 0$.

We estimated confidence interval for tau as following: since $\tau$ depends on $b$, its standard error is derived using error propagation:

$$\sigma_\tau = \frac{\sigma_b}{b^2}, \tag{17}$$

where $\sigma_b$ is the standard error of $b$, obtained from the covariance matrix of the fitted parameters.

To compute the 95% confidence interval for $\tau$, we use the Student's t-distribution, which accounts for small sample sizes:

$$\tau_{\mathrm{lower}} = \tau - t_{\alpha/2,\mathrm{dof}} \cdot \sigma_\tau, \tag{18}$$

$$\tau_{\mathrm{upper}} = \tau + t_{\alpha/2,\mathrm{dof}} \cdot \sigma_\tau, \tag{19}$$

where:

- $t_{\alpha/2,\mathrm{dof}}$ is the critical t-value for a 95% confidence level,

- dof is the degrees of freedom (number of data points − number of parameters).

## 4.4 Intrinsic timescale estimation in the synthetic dataset

Intrinsic timescales were estimated at the level of individual units using four methods. The autocorrelation function was computed for all units without any exclusion criteria. If the computation failed at a specific time lag, the corresponding value was set to NaN.

For iSTTC, the following parameters were used: number of lags $n_{lags}$ = 20, *lag_shift* = 50 ms and $\Delta t$ = 25 ms. When applying iSTTC to trial-based data, we additionally used a *zero_padding_len* = 3000 ms, which corresponds to the amount of zero-padding appended to each concatenated trial. For ACF and PearsonR, spike trains were binned with a bin size of 50 ms, and the number of lags was set to $n_{lags}$ = 20.

The intrinsic timescale was determined by fitting a single-exponential decay function to the autocorrelation curve, starting from a time lag of 1. Autocorrelation functions containing NaN values were excluded from the fitting process.

## 4.5 Intrinsic timescale estimation in the mouse dataset

Intrinsic timescales were estimated at both the single-unit and brain-area levels using the same four methods. The autocorrelation function was computed for all units without any exclusion criteria. If the computation failed at a specific time lag, the corresponding value was set to NaN.

The same parameter settings were used as in the synthetic dataset.

For single units, the intrinsic timescale was determined by fitting a single-exponential decay function to the autocorrelation curve, starting from a time lag of 1. For brain areas, the fit was performed starting from a time lag of 2. Autocorrelation functions containing NaN values were excluded from the fitting process. To estimate the area-level intrinsic timescale, all valid ACFs (those without NaNs) from the area were used to fit a single exponential function.

## 4.6 Intrinsic timescale estimation with adaptive Approximate Bayesian Computations

To estimate ITs with aABC we used the python package abcTau ([29], https://github.com/roxana-zeraati/abcTau).

As generative model, we used the doubly stochastic process with a single intrinsic timescale and gamma-distributed spike counts (generativeModel = "oneTauOU_gammaSpikes"). As the summary statistic, we used the autocorrelation computed in the time domain (summStat_metric = "comp_cc"), and for the distance metric we used the linear distance (distFunc = "linear_distance").

The initial error threshold was set to $\epsilon$ = 1 for all fits. aABC iterations proceeded until the acceptance rate decreased to accR $\leq$ 0.01. We required a minimum of 50 accepted samples for the posterior distribution. As the IT estimate for each unit, we took the maximum a posteriori (MAP) value of the inferred timescale parameter. We used a uniform prior over the timescale parameter, ranging from 0 to 400 ms.

The fitting procedure was subject to a per-unit time limit. Based on the median runtime of 3.5 minutes per fit, we terminated any run that exceeded 15 minutes during the first iteration. Units that did not converge within this limit were considered not estimated and were marked as rejected.

## 4.7 Local variation

Local variation *Lv* was computed as follows [38]:

$$Lv = \frac{3}{n-1} \sum_{i=1}^{n-1} \left( \frac{I_i - I_{i+1}}{I_i + I_{i+1}} \right)^2,$$

(20)

where $I_i$ and $I_{i+1}$ are the *i*-th and *i*+1st ISIs, and n is the number of ISIs.

Lv equals 0 for regular spike trains, 1 for Poisson spike trains and is above 1 for bursty spike trains.

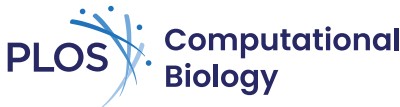

## 4.8 Relative estimation error

Relative estimation error *REE* was computed for the synthetic dataset as follows:

$$REE = \frac{IT_e - IT_{gt}}{IT_{gt}} \times 100,$$

(21)

where $IT_e$ is an estimated intrinsic timescale and $IT_{gt}$ is the ground truth intrinsic timescale.

To compute the REE of PearsonR and iSTTC trials in the mice dataset, we used as ground truth the intrinsic timescale estimated using ACF or iSTTC, respectively. For the REE of methods applied to spike trains of varying lengths or different numbers of trials, the intrinsic timescale computed from the full signal was used as ground truth.

## 4.9 Computation time estimation

To evaluate the computational efficiency and scalability of the different intrinsic timescale estimation methods, we measured the wall-clock time required to compute the autocorrelation function and to fit the exponential decay model for each unit. For aABC, we quantified the total computation time per unit, with all computations performed without parallelisation. All computations were performed on the same hardware (Windows 11 Pro, Intel Core i7-8700 CPU, 32 GB RAM (HP ProDesk 400 G6 MT)).

## 4.10 Statistical analysis

Statistical analysis was done in R. Non-nested data were analyzed with (generalized) linear models and nested data with (generalized) linear mixed-effects models. Initial model selection was based on exploratory analysis of the data, and then the goodness of fit was evaluated using explained variance (R-squared) and residuals distribution (check_model function from performance R package). When several models were fit on the same data, the model fit was compared using compare_performance function from performance R package. 95% confidence intervals were computed using confint R function, p-values for linear mixed-effects models were computed with the lmerTest R package, post hoc analysis with Tukey multiple comparison corrections was done using emmeans R package, and the model fit was plotted using sjPlot and ggplot2 R packages.

## 4.11 Code availability

The complete analysis pipeline, including synthetic dataset and figures generation, is available at the following open-access repository: https://github.com/iinnpp/isttc.

## Supporting information

**S1 Fig. Representative examples of autocorrelation functions and exponential fits for units where iSTTC or ACF performs better on unsegmented spike trains. (A)** Units with high excitation strength and low firing rates, where iSTTC provides more accurate fits. Units with longer intrinsic timescales are shown on the left and shorter timescales on the right. **(B)** Units with low excitation strength and high firing rates, where ACF performs better. Units with longer intrinsic timescales are shown on the left and shorter timescales on the right. In **(A)** and **(B)**, the grey or blue scatter shows the unit's estimated autocorrelation function, the corresponding grey or blue solid line shows the fitted exponential decay function, and the black line shows the ground-truth autocorrelation function of the underlying Hawkes process. In **(A)** and **(B)**, for visualization purposes, the ground-truth autocorrelation is scaled separately in each panel to match the amplitude of the corresponding empirical estimate at the first non-zero lag, i.e., $ACF_{true,scaled}(t) = \frac{ACF_{emp}(lag\_1)}{ACF_{true}(lag\_1)} ACF_{true}(t)$, where $lag\_1$ denotes the first non-zero lag.
(TIF)

**S2 Fig. Local variation is an experimental proxy for the excitation strength parameter of the Hawkes point process. (A)** Effect plot of the statistical model used to investigate the REEs plotted in Fig 3C–3D. Negative values indicate reduced REE. ACF is used as the reference method. The x symbol denotes an interaction term between factors. **(B)** Schematic representation of local variation (Lv), modified from [38], and examples of spike trains with regular, random, and bursty firing patterns. Dice icon from svgrepo.com. **(C)** Effect plot of the statistical model used to investigate how Lv correlates with the Hawkes point parameters. **(D)** KDE plot displaying the distribution of Lv in the synthetic dataset, and, in the inset, the line plot displaying predicted Lv values across excitation strength (left). Line plot displaying predicted REE values for iSTTC and ACF across Lv (n = $10^5$ single units) (right). Shaded areas represent 95% confidence intervals. Y-axes are plotted on a $log_{10}$ scale. In **(A)** and **(C)**, data is presented as mean difference (blue dots) with 95% confidence interval. In **(A)**, **(C)**, and **(D)**, asterisks indicate a significant effect. * $p < 0.05$, *** $p < 0.001$. Generalized linear models with interactions **(A)** and **(D)**. Generalized linear model **(C)**. (TIF)

**S3 Fig. Computational efficiency and scalability of iSTTC and ACF on unsegmented spike trains. (A)** Violin plot displaying per-unit computation time required to compute the autocorrelation function (left), exponential fit (middle), and total per-unit computation time (right). **(B)** Effect plot of the statistical model used to investigate how computation time correlates with method, firing rate, excitation strength, and intrinsic timescale. ACF is used as the reference method. The x symbol denotes an interaction term between factors. **(C)** Scatter plot displaying the total computation time as a function of dataset size. **(D)** Line plot displaying predicted computation time values for iSTTC and ACF as a function of signal length (left), and effect plot of the statistical model used to investigate how computation time correlates with signal length (right). Shaded areas represent 95% confidence intervals. In **(A)**, data is presented as median, 25th, 75th percentile, and interquartile range, with the shaded area representing the probability density distribution of the variable. In **(A)**, asterisks indicate a significant effect of the IT estimation method. In **(D)** left, asterisks indicate a significant effect of an interaction between the IT estimation method and signal length. In **(A)**, **(B)** and **(D)**, asterisks indicate a significant effect. *** $p < 0.001$. Generalized linear model with interactions **(A)**, **(B)** and **(D)**. (TIF)

**S4 Fig. iSTTC yields significantly lower REEs than PearsonR on epoched data. (A)** Effect plot of the statistical model used to investigate the REEs plotted in Fig 4B–4C. Negative values indicate reduced REE, and PearsonR is used as the reference method. The x symbol denotes interaction terms between factors. In **(A)**, data is presented as mean differences (blue dots) with 95% confidence intervals. In **(A)**, asterisks indicate a significant effect. * $p < 0.05$, ** $p < 0.01$, *** $p < 0.001$. Generalized linear model with interactions **(A)**. (TIF)

**S5 Fig. Representative examples of autocorrelation functions and exponential fits for units where iSTTC or PearsonR performs better on epoched spike trains. (A)** Units with high excitation strength, where iSTTC provides more accurate fits. Units with longer intrinsic timescales are shown on the left and shorter timescales on the right. **(B)** Units with low excitation strength, where PearsonR performs better. Units with longer intrinsic timescales are shown on the left and shorter timescales on the right. In **(A)** and **(B)**, the orange or violet scatter shows the unit's estimated autocorrelation function, the corresponding orange or violet solid line shows the fitted exponential decay function, and the black line shows the ground-truth autocorrelation function of the underlying Hawkes process. In **(A)** and **(B)**, for visualization purposes, the ground-truth autocorrelation is scaled separately in each panel to match the amplitude of the corresponding empirical estimate at the first non-zero lag, i.e., $ACF_{true,scaled}(t) = \frac{ACF_{emp}(lag\_1)}{ACF_{true}(lag\_1)} ACF_{true}(t)$, where $lag\_1$ denotes the first non-zero lag. (TIF)

**S6 Fig. Computational efficiency and scalability of iSTTC and PearsonR on epoched spike trains. (A)** Violin plot displaying per-unit computation time required to compute the autocorrelation function (left), exponential fit (middle), and

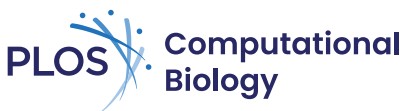

total per-unit computation time (right). **(B)** Effect plot of the statistical model used to investigate how computation time correlates with method, firing rate, excitation strength, and intrinsic timescale. PearsonR is used as the reference method. The x symbol denotes an interaction term between factors. **(C)** Scatter plot displaying the total computation time as a function of dataset size. **(D)** Line plot displaying predicted computation time values for iSTTC and PearsonR as a function of the number of trials (left), and effect plot of the statistical model used to investigate how computation time correlates with the number of trials (right). Shaded areas represent 95% confidence intervals. In **(A)**, data is presented as median, 25th, 75th percentile, and interquartile range, with the shaded area representing the probability density distribution of the variable. In **(A)**, asterisks indicate a significant effect of the IT estimation method. In **(D)** left, asterisks indicate a significant effect of an interaction between the IT estimation method and number of trials. In **(A)**, **(B)** and **(D)**, asterisks indicate a significant effect. * $p < 0.05$, ** $p < 0.01$, *** $p < 0.001$. Generalized linear model with interactions **(A)**, **(B)** and **(D)**. (TIF)

**S7 Fig. Epoched and short unsegmented data lead to unreliable IT estimates. (A)** Effect plot of the statistical model used to investigate the REEs plotted in Fig 5B. Negative values indicate reduced REE, and ACF is used as the reference method. The x symbol denotes interaction terms between factors. **(B)** Same as **(A)** for the REEs plotted in Fig 5D. **(C)** Line plot showing the percentage of spike trains whose standard error of the median IT estimate falls within a specified tolerance range (5% to 15%) across resampling iterations. **(D)** Line plot displaying the IT estimate standard error of the median as a function of the number of resampling iterations. In **(A)** and **(B)**, data is presented as mean differences (blue dots) with 95% confidence intervals. In **(D)**, data is presented as mean with 95% confidence intervals. In **(A)** and **(B)**, asterisks indicate a significant effect. * $p < 0.05$, *** $p < 0.001$. Generalized linear model with interactions **(A)** and **(B)**. (TIF)

**S8 Fig. Sensitivity of IT estimates to hyperparameter choices. (A)** Left, violin plots showing the difference between the estimated intrinsic timescales and the reference intrinsic timescales as a function of bin size (ACF) or lag shift (iSTTC). Estimates obtained with a 50-ms bin size (lag shift) serve as the reference. Right, effect plot of the statistical model used to investigate how bin size (lag shift) and method interact to influence intrinsic timescale estimates. ACF is used as the reference method. The x symbol denotes an interaction term between factors. **(B)**, same as **(A)** for PearsonR and iSTTC on epoched data. In **(A)** left and **(B)** left, data is presented as median, 25th, 75th percentile, and interquartile range, with the shaded area representing the probability density distribution of the variable. In **(A)** right and **(B)** right, asterisks indicate a significant effect. * $p < 0.05$, ** $p < 0.01$, *** $p < 0.001$. Generalized linear model with interactions **(A)** and **(B)**. (TIF)

**S9 Fig. Representative examples illustrating how different methods of autocorrelation function estimation lead to different inclusion outcomes. (A)** Representative examples of autocorrelation functions and exponential fits for units. For each example unit, the colored scatter shows the estimated autocorrelation function, the corresponding colored line shows the fitted exponential decay function, and the black line shows the ground-truth autocorrelation function of the Hawkes process. The top row shows a unit for which all methods satisfy the inclusion criteria. The middle row shows a unit where trial-based methods begin to fail the criteria, although the ACF still exhibits a clear exponential decline. The bottom row shows a unit that violates all inclusion criteria for the trial-based methods. For visualization purposes, the ground-truth autocorrelation is scaled separately in each panel to match the amplitude of the corresponding empirical estimate at the first non-zero lag, i.e., $ACF_{true,scaled}(t) = \frac{ACF_{emp}(lag\_1)}{ACF_{true}(lag\_1)} ACF_{true}(t)$, where $lag\_1$ denotes the first non-zero lag. (TIF)

**S10 Fig. iSTTC allows inclusion of more units than PearsonR or ACF, especially for low trial counts and short signals. (A)** Scatter plots showing the percentage of units across increasing signal lengths for ACF and iSTTC. From left to right: percentage of excluded fits, percentage of units with autocorrelation function decline in the 50–200 ms range,

percentage of units with 95% confidence intervals excluding zero, and percentage of units with R-squared $\geq$ 0.5. **(B)** Same as **(A)**, but for PearsonR and iSTTC across increasing numbers of trials. **(C)** Scatter plot showing the percentage of excluded units as a function of bin size (ACF and PearsonR) or lag shift (iSTTC). Left, percentage of excluded units for iSTTC and ACF, right, for iSTTC on trials and PearsonR.
(TIF)

**S11 Fig. Comparison of abcTau with other IT estimation methods. (A)** Comparison of IT estimation accuracy across five methods. Ridgeline plot displaying the distribution of REE. Only IT estimates with REE between 0% and 100% are shown. Percentages indicate the proportion of IT estimates within this interval for each method (left). Scatter plot displaying the percentage of IT estimates with REE falling within progressively narrower intervals (middle). Violin plot displaying the full distribution of REE values for each method. (right). **(B)** Hexbin plot displaying the difference in REE between iSTTC and abcTau as a function of firing rate and excitation strength (left), and IT and excitation strength (right) (n = 10000 single units, 40 trials x 1000 ms each). Color codes for the median REE difference in each bin, with blue indicating lower error for iSTTC. **(C)** Left, bar plot displaying the percentage of excluded units across five methods. Right, kernel density plot displaying the distribution of excluded units as a function of firing rate (left), excitation strength (middle), and IT (right). **(D)** Violin plot displaying total per-unit computation time (left) and scatter plot displaying the total computation time as a function of dataset size. In **(A) right** and **(D) left**, data is presented as median, 25th, 75th percentile, and interquartile range, with the shaded area representing the probability density distribution of the variable.
(TIF)

**S12 Fig. Trial-based IT estimation methods yield unreliable IT estimates on an experimental dataset. (A)** Bar plot displaying the number of units per brain area (left), violin plot displaying the firing rate per brain area (middle), and violin plot displaying the Lv per brain area (right). The grey line denotes the Lv = 1 that corresponds to a random firing pattern. **(B)** Scatter plot displaying brain area level ITs estimated by four methods across individual brain areas. **(C)** Heatmap displaying brain area level ITs estimated by four methods across individual brain areas. Color codes for the estimated IT, with warmer colors indicating higher ITs. **(D)** Scatter plot displaying cortical brain area level ITs estimated by four methods as a function of anatomical hierarchy score [2,36]. **(E)** Effect plot of the statistical model used to investigate the ITs plotted in Fig 7C, left. Negative values indicate reduced pseudo-REE, and ACF is used as the reference method. The x symbol denotes interaction terms between factors. **(F)** Same as **(E)** for the pseudo-REEs plotted in Fig 7C, middle. In **(A)**, data is presented as median, 25th, 75th percentile, and interquartile range, with the shaded area representing the probability density distribution of the variable. In **(B)** and **(D)**, data is presented with 95% confidence intervals. In **(E)** and **(F)**, data is presented as mean differences (blue dots) with 95% confidence intervals. In **(E)** and **(F)**, asterisks indicate a significant effect. ** $p < 0.01$, *** $p < 0.001$. Generalized linear model with interactions **(E)** and **(F)**.
(TIF)

**S13 Fig. iSTTC outperforms ACF and PearsonR on short unsegmented and epoched data. (A)** Effect plot of the statistical model used to investigate the pseudo-REEs plotted in Fig 7D. Negative values indicate reduced pseudo-REE, and ACF is used as the reference method. The x symbol denotes interaction terms between factors. **(B)** Same as **(A)** for the pseudo-REEs plotted in Fig 7F. **(C)** Line plot showing the percentage of spike trains whose standard error of the median IT estimate falls within a specified tolerance range (5% to 15%) across resampling iterations. **(D)** Line plot displaying the IT estimate standard error of the median as a function of the number of resampling iterations. In **(A)** and **(B)**, data is presented as mean differences (blue dots) with 95% confidence intervals. In **(D)**, data is presented as mean with 95% confidence intervals. In **(A)** and **(B)**, asterisks indicate a significant effect. *** $p < 0.001$. Generalized linear model with interactions **(A)** and **(B)**.
(TIF)

**S14 Fig. iSTTC allows inclusion of more units than PearsonR also on an experimental dataset. (A)** Bar plot displaying the percentage of excluded units across four methods. Color codes indicate exclusion reasons, with dark grey indicating failed exponential fits and light grey indicating negative R-squared values of the exponential fit (n = 6481 excluded fits across four methods, n = 1214 excluded fits ACF, n = 1290 excluded fits iSTTC, n = 2345 excluded fits PearsonR, n = 1632 excluded fits iSTTC (trials)). **(B)** Kernel density plot displaying the distribution of excluded units as a function of firing rate (left) and local variation Lv (right) (n = 1412 units both methods failed, n = 933 units only PearsonR, n = 220 units only iSTTC). **(C)** Bar plots displaying the percentage of units with autocorrelation function decline in the 50–200 ms range (left, n = 9331 fits across four methods), and the percentage of units with 95% confidence intervals excluding zero (right, n = 9146 fits across four methods), across four methods. **(D)** Kernel density plot displaying the distribution of R-squared values (left), and bar plot displaying the percentage of units with R-squared $\geq$ 0.5 across four methods (right, n = 12102 fits across four methods).
(TIF)

## Author contributions

**Conceptualization:** Irina Pochinok, Ileana L. Hanganu-Opatz, Mattia Chini.

**Data curation:** Irina Pochinok.

**Formal analysis:** Irina Pochinok.

**Funding acquisition:** Ileana L. Hanganu-Opatz.

**Investigation:** Irina Pochinok.

**Methodology:** Irina Pochinok, Mattia Chini.

**Project administration:** Ileana L. Hanganu-Opatz, Mattia Chini.

**Resources:** Ileana L. Hanganu-Opatz.

**Software:** Irina Pochinok.

**Supervision:** Ileana L. Hanganu-Opatz, Mattia Chini.

**Validation:** Mattia Chini.

**Visualization:** Irina Pochinok.

**Writing – original draft:** Irina Pochinok.

**Writing – review & editing:** Irina Pochinok, Ileana L. Hanganu-Opatz, Mattia Chini.

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
