## [Decision Letter · Decision Letter 0]

5 Oct 2025

iSTTC: a robust method for accurate estimation of intrinsic neural timescales from single-unit recordings

PLOS Computational Biology

Dear Dr. Pochinok,

Thank you for submitting your manuscript to PLOS Computational Biology. After careful consideration, we feel that it has merit but does not fully meet PLOS Computational Biology's publication criteria as it currently stands. Therefore, we invite you to submit a revised version of the manuscript that addresses the points raised during the review process.

Please submit your revised manuscript within 60 days Dec 05 2025 11:59PM. If you will need more time than this to complete your revisions, please reply to this message or contact the journal office at ploscompbiol@plos.org. Please include the following items when submitting your revised manuscript:

We look forward to receiving your revised manuscript.

Kind regards,

Sacha Jennifer van Albada

Academic Editor

PLOS Computational Biology

Daniele Marinazzo

Section Editor

PLOS Computational Biology

**Journal Requirements:**

4) We notice that your supplementary Figures are included in the manuscript file. Please remove them and upload them with the file type 'Supporting Information'. Please ensure that each Supporting Information file has a legend listed in the manuscript after the references list.

Potential Copyright Issues:

i) Figures 2A, and 6A. Please confirm whether you drew the images / clip-art within the figure panels by hand. If you did not draw the images, please provide (a) a link to the source of the images or icons and their license / terms of use; or (b) written permission from the copyright holder to publish the images or icons under our CC BY 4.0 license. Alternatively, you may replace the images with open source alternatives. See these open source resources you may use to replace images / clip-art:

6) Thank you for stating that "Synthetic dataset is available at https://gin.g-node.org/iinnpp/isttc.

This link reaches a 404 error page. Please amend this to a new link or provide further details to locate the data.

**Reviewers' comments:**

Reviewer's Responses to Questions

Reviewer #1: In this manuscript, the authors propose a new method (iSTTC) to estimate the auto-correlation time from spiking trains of single neurons. This auto-correlation time, frequently called 'intrinsic timescale' (IT), has been of continuing interest because it varies systematically across brain regions and tends to be correlated with behavior. Accordingly, the method is potentially relevant for the neuroscience community. Furthermore, it is, to the best of my knowledge, novel.

The manuscript starts with a concise but nicely executed summary of the ongoing interest in intrinsic timescales. In the results, the authors start with a brief description of their method (2.1), then they compare it to existing methods on simulated data (2.2-2.5), and finally they apply their method to mouse data from the Allen Institute (2.6). Structure, writing, and figures are mostly clear and easy to follow. Commendably, the code is publicly accessible and clearly organized; it runs but I did not check the implementation in detail.

The main result is that the proposed method outperforms existing ones. While the authors demonstrate this for two commonly used approaches, my main criticism is that they did not compare it to the method by Zeraati, Engel, and Levina (ref 25 in the manuscript). The latter already addresses several of the problems discussed in the third paragraph of the introduction, accordingly it provides a much more competitive baseline. Taking also the remarks below into account, I believe that the manuscript needs a major revision.

Major remarks:

* The dominant effect in the Figures seems to be that all methods have an estimation error of at least 10 (often 100, sometimes 1000) percent. Furthermore, the total error is always much larger than the difference between iSTTC and the compared method. Naively, this seems to suggest that there is currently no method to reliably estimate the IT - which makes it even more important to compare to ref. 25.

* It is not clear to me why spike trains can be concatenated in iSTTC (claimed several times to be "a key advantage"), and I did not find an explanation for this in the Methods.

* Regarding the ground-truth model: Is the auto-correlation function for this model exponential? And is tau = tau_kernel / (1 - alpha) the time scale of this exponential (the parameters seem to be set based on this assumption)?

* Given that the manuscript introduces the method it would be nice to provide a detailed explanation as well as a graphical overview (similar to Supp. Sec. 7.2.2. and Supp. Fig. 1) in the main text. Conversely, the description in 2.1 is too brief to understand the method.

Minor remarks:

* "Ogata’s thinning algorithm" needs a reference.

* Eq. (2) seems to contain typos: it should be g(Delta) instead of gDelta, there is a trailing bracket, there should be a comma before Delta ge 0.

* Below eq (2) I believe tau -> tau_kernel below eq (2).

* Missing dependencies in the repository: jupyter, matplotlib, statsmodels, seaborn

* The link to gin for the synthetic dataset does not work (404 error).

* In Fig. 6: should "local variance" be "local variation"?

Reviewer #2: This manuscript introduces intrinsic Spike Time Tiling Coefficient (iSTTC), an extension of the spike-time tiling coefficient designed to provide more robust and inclusive estimates of intrinsic neural timescales (INTs) from single-unit recordings. The study is timely because INTs are increasingly recognized as fundamental markers of neural computation and cognition, yet current approaches remain limited by strict inclusion criteria, biases in trial-based data, and instability in the presence of sparse firing. By introducing a method that works well across both continuous and epoched recordings, the authors make an important methodological contribution. The approach is mostly solid, combining synthetic datasets with known ground truth and large-scale Neuropixels recordings, and the authors also demonstrate improvements in inclusion rates, reliability, and robustness. In my opinion, this is a solid contribution that may help future studies of temporal integration in neural systems.

Major Points

1. A fundamental, yet only tacitly acknowledged, limitation in the field is the absence of a definitive ground truth for identifying interneurons (INTs) in experimental data. While synthetic models permit validation, this is inherently impossible for in vivo recordings. The manuscript proceeds without directly addressing this core issue, presenting results as if a ground truth were available. We strongly recommend that the Discussion explicitly confronts this limitation, highlighting the shortcomings and possible ways to mitigate them.

2. The enhanced inclusion rates provided by iSTTC are a valuable advancement. However, it is equally important to consider the neurons that remain excluded. It is unclear whether these units constitute a distinct biological subclass (e.g., neurons with extremely low firing rates or high irregularity), whose systematic omission could bias population-level inferences. A deeper discussion is needed on whether these neurons should be accounted for and what the biological implications of their inclusion or exclusion truly are.

3. The manuscript's comparative analysis is currently limited to ACF and Pearson R. To more fully establish the utility of iSTTC, we recommend contextualizing it against other prominent methods for assessing temporal structure. These include, but are not limited to, the autocorrelation window, detrended fluctuation analysis, Hurst exponent, and measures of spectral slope. Even a conceptual discussion of iSTTC's performance relative to these approaches would help clarify its distinct advantages and limitations within the existing methodological ecosystem.

4. The reliance on a fixed 50 ms lag parameter lacks rigorous justification. The manuscript would be significantly strengthened by a systematic exploration of how iSTTC's performance and the resulting inclusion rates vary across a spectrum of lag values. Demonstrating robustness across timescales (e.g., from fine 10 ms to coarser 100 ms windows) is critical for validating the method and interpreting the biological meaning of the selected integration window.

5. While the experimental validation may effectively recapitulate the known hierarchy of timescales, the results should present this in a more direct form with the regions ordered in the x-axis by the hierarchucal index. Moreover, the Discussion could more deeply integrate this result with the anatomical and physiological principles that are thought to underlie it, especially given contradictory reports in the literature.

Siegle JH, Jia X, Durand S, Gale S, Bennett C, Graddis N, et al. Survey of spiking in the mouse visual system reveals functional hierarchy. Nature. 2021;592(7852):86–92.

Piasini E, Soltuzu L, Muratore P, Caramellino R, Vinken K, Op de Beeck H, et al. Temporal stability of stimulus representation increases along rodent visual cortical hierarchies. Nature Communications. 2021;12(1):4448.

This contradiction has also been extended more broaddly in the literaure in which peripheral regions have been proposed to be closer to criticality

https://www.pnas.org/doi/full/10.1073/pnas.2208998120

or farther from criticality

https://journals.aps.org/prx/abstract/10.1103/PhysRevX.14.031021

In general, explicitly connecting iSTTC-derived timescales to features like cortico-cortical connectivity profiles, laminar-specific circuitry, and mechanisms of dendritic integration would firmly embed the method within a dominant theoretical framework and suggest specific biological mechanisms for the observed differences.

6. The simulations convincingly show that firing rate affects estimation error, but this could be more deeply contextualized in relation to previous work showing that firing rate and dendritic morphology modulate timescales mechanistically. Integrating these findings could reinforce the idea that timescales reflect both statistical and physiological processes.

https://doi.org/10.3389/fncel.2024.1404605

Minor Points:

1. Guidance for Trial-Based Designs: The paper effectively critiques trial-based approaches but does not provide practical guidance for experimentalists. To maximize its utility, the authors should discuss the conditions under which iSTTC remains reliable in a trial-based setting (e.g., minimum number of trials, trial length, and firing rate thresholds) and clarify when continuous recordings are truly essential.

2. Capturing Multi-Scale Dynamics: The analysis assumes neuronal timescales are well-described by a single exponential. It is unclear whether iSTTC can adequately capture the dynamics of neurons with multiple intrinsic timescales. The validity of this assumption should be addressed, potentially by testing alternative models (e.g., double exponentials or power laws) on suitable data.

3. Computational Efficiency and Scalability: A comment on the computational efficiency of iSTTC is warranted. For the method to be widely adopted, it is important to demonstrate its scalability to very large datasets (e.g., high-density Neuropixels recordings) without prohibitive computational cost. Please include information on algorithmic complexity and computation time.

4. Biological Mechanisms in Discussion: The Discussion would be strengthened by more explicitly framing intrinsic timescales (INTs) not just as statistical constructs but as manifestations of underlying biological mechanisms. Briefly linking the findings to substrates like NMDA receptor kinetics, synaptic time constants, and circuit-level excitation-inhibition balance would better bridge the methodological advance to its physiological relevance.

5. Terminology Consistency: The terms “ACF,” “ACF-like,” and “pseudo-autocorrelations” should be used consistently to avoid reader confusion. We recommend standardizing all terms throughout the manuscript.

Reviewer #3: The manuscript introduces a new method, called iSTTC, to estimate the ACF of single-unit activity reliably. iSTTC is an extension of the previously developed spike time tilling coefficient method used for computing correlations between two different spike trains. The authors did a great job of comparing iSTTC to previous methods on synthetic data with known ground truth, as well as on experimental data. The manuscript is clearly written, and the proposed method could provide significant improvements for estimating timescales from single-unit activity. I have just some comments that may help clarify some aspects of the method and its sensitivity to hyperparameters.

Major comments:

1. Since the main result of this paper is the proposed method itself, it would be helpful to readers to describe the general steps of the method already in the first section of the Results, before discussing how it compares to previous methods. Moreover, a figure illustrating how iSSTC works could clarify some of the comments I am raising below.

2. Some details of the method are not clear from the current manuscript. I hope the comments below help to clarify them:

- Authors discuss that iSTTC works without needing to bin the data, but in section 7.2.2 they mention n_lags parameter (the total number of time lags), which seems to be an integer. Without binning, the lags should be computed in continuous time, not as an integer, so it was unclear how this parameter is computed and what its role is.

- For the trialed version of iSTTC, how is the length of zero padding selected? For experimental data, should it match the inter-epoch intervals of the actual epoched data? Also, is iSTTC computed between spikes across trials (I think here a figure illustration could be quite helpful).

- In Methods, the authors mention “To estimate the area-level intrinsic timescale, all valid ACFs (those without NaNs) from the area were used to fit a single exponential function.” For this, did the authors average over ACF all neurons and then fit the average ACF, or did they fit a single function to all individual ACFs? In the latter, what is the logic behind having the same timescale for all neurons, given that from many studies we know there is a large diversity across single-neuron timescales within an area?

- It would be helpful if the authors show example ACFs and their exponential fits from both methods in Figs 2 and 3, in addition to statistics, to visualize cases where each method is better and also compare them to the ground-truth shape of ACF of the Hawkes process. Also, for Fig 5, it would be helpful to see how the shape of example ACFs changes depending on the method and highlight how it affects their inclusion based on different criteria.

3. The authors discuss that iSTTC method has several hyperparameters, such as n_lags, lag_shift, Delta t, and zero_padding length. For the application of the method, it would be important to show how the choice of these hyperparameters affects the estimated timescale. Testing for hyperparameter-sensitivity is also important for comparing with classic methods that use bin size as a hyperparameter. The selected hyperparameters need to be mentioned in the figure captions (or as a table).

Minor comments:

- This sentence, “In conclusion, also on experimental data, iSTTC outperforms ACF and PearsonR IT estimation accuracy.”, is not formulated quite accurately, since for the experimental data, there is no ground truth to test which method outperforms which.

- The authors use “continuous data” to refer to long trials/time-series, but the term is a bit misleading, especially in the abstract, since it might convey that iSTTC can be applied to continuous signals such as LFP.

**Have the authors made all data and (if applicable) computational code underlying the findings in their manuscript fully available?**

Reviewer #1: **No:** Synthetic data not available

Reviewer #2: Yes

Reviewer #3: Yes

PLOS authors have the option to publish the peer review history of their article (what does this mean? ). If published, this will include your full peer review and any attached files.

**Do you want your identity to be public for this peer review?** For information about this choice, including consent withdrawal, please see our Privacy Policy .

Reviewer #1: No

Reviewer #2: No

Reviewer #3: No

**Figure resubmission:**
---

## [Decision Letter · Decision Letter 1]

3 Feb 2026

PCOMPBIOL-D-25-01542R1

iSTTC: a robust method for accurate estimation of intrinsic neural timescales from single-unit recordings

PLOS Computational Biology

Dear Dr. Pochinok,

Thank you for submitting your manuscript to PLOS Computational Biology. After careful consideration, we feel that it has merit but does not fully meet PLOS Computational Biology's publication criteria as it currently stands. Therefore, we invite you to submit a revised version of the manuscript that addresses the points raised during the review process.

We look forward to receiving your revised manuscript.

Kind regards,

Sacha Jennifer van Albada

Academic Editor

PLOS Computational Biology

Daniele Marinazzo

Section Editor

PLOS Computational Biology

**Journal Requirements:**

**Reviewers' comments:**

Reviewer's Responses to Questions

**Comments to the Authors:**

Reviewer #1: I thank the authors for their detailed response. In combination with the changes in the manuscript, all my concerns are very well addressed. I recommend publication after addressing the minor comments (typos etc.) below.

In my opinion, the revision substantially improved the manuscript. In particular, the extended introduction of the method in Section 2.1 and Figure 2 make it much easier to understand the method - including the implementation of trial concatenation. The detailed comparison to adaptive Approximate Bayesian Computation roots the work nicely in the literature (or, as another reviewer put it, the methodological ecosystem) and highlights its strengths. Finally, I very much appreciate the additional discussions regarding fundamental limits in the estimation of intrinsic timescales.

Minor comments:

* Typesetting in Eq. (1): Delta and t should both be in the subscript

* Fig. 2 caption: what is t (presumably a Delta is missing)?

* The supplemental figures in the zip file seem mixed up (e.g., Supp. Fig. 11 seems to be SuppFigure8.tiff)

Beyond the scope of this review process, I am very curious if there is a direct mathematical link between iSTTC and the more traditional ACF estimates. For example, do they converge to the same value for infinite data? If there is such a link, adding it to the supplement would in my opinion significantly strengthen the underlying theoretical foundations of the method (and also provide some insight on the unification of measures discussed in response to reviewer 2, point 3). However, in my opinion it should be fully at the authors' discretion whether or not to include this; the empirical evidence in the manuscript provides a clear picture.

Reviewer #2: All my concerns have been addressed, and I recommend publication.

Reviewer #3: I thank the authors for considering my comments and carefully addressing them, which improved the clarity of the manuscript. I have a few remaining comments on the revised manuscript:

1. I am confused by Supp. Fig 1 and 5, how come the ground-truth autocorrelation (black line) is sometimes different between iSTTC and ACF panels (e.g., the first row of panel A in Supp. Fig 1)? The ground-truth autocorrelation should be independent of how the data autocorrelation is estimated (by iSTTC or ACF).

2. The filenames of some of the attached tif files of supplementary figures do not match their captions/descriptions in the text. For instance, the Supp. Fig 9 tif file is Supp. Fig 8, Supp. Fig 10 tif file is probably Supp. Fig 11, etc. At least for reviewing, it would have been helpful if the supplementary figures were included in the manuscript together with their captions.

3. A few comments about the new section comparing aABC and iSTTC:

- The authors mentioned (line 338) “aABC is a method that improves IT estimation by optimizing the exponential fitting of the ACF”; this is not accurate because the aABC method does not fit exponential functions at all; it replaces the exponential fits by comparing summary statistics of synthetic data from a generative model with the real data.

- I completely agree with the authors that these two methods are addressing two complementary problems, ACF estimation versus timescale estimation, and in principle, their combination could improve the overall process. At the same time, choosing the right summary statistics is an important step for all simulation-based inference methods, including ABC or SBI; therefore, it is not quite right to compare aABC with iSTTC when the summary statistics are not matched. Considering this point, it is misleading to suggest aABC is not suitable for SUA data because of its higher rejection rate. If authors use iSTTC as the summary statistic of aABC, the rejection rates will probably be comparable, but I agree that this is beyond the scope of this manuscript. My suggestion is that authors revise this section to clarify the difference in the summary statistics between methods and the fact that with better summary statistics, aABC will probably have comparable results for epoched SUA data.

- Of course, it is fair to say that aABC method has a higher computational cost because it uses simulations from generative models instead of exponential fitting. However, again, the speed of the simulation-based methods depends on the chosen summary statistics and the generative model. Based on the toolbox description (https://github.com/roxana-zeraati/abcTau), it seems the authors have chosen the slowest summary statistics “comp_cc” which computes Pearson correlations in the time domain, but presumably this step could be significantly sped up if the authors used an FFT-based method (Although it would still be slower than exponential fitting, the differences may be smaller). This point needs to be clarified in the manuscript.

- and of course, as the authors mentioned, for long time series, aABC does not add much given that the ACF bias is negligible. So it does not make sense to use aABC for unsegmented data with long durations if there is no bias.

4. Fig 2C caption needs a bit more detailed description. For example, how is the intrinsic timescale difference calculated? Is the difference from the ground truth? If yes, the parameters of the process/ground truth need to be specified.

5. There is a typo in equation 1: it should be STTC_{\Delta t}

6. There is a mismatch between the notation that is used in Figure 2, its caption, and the main text description. The time lag in the main text and figure is \Delta t, but in the figure caption, it is t.

**Have the authors made all data and (if applicable) computational code underlying the findings in their manuscript fully available?**

Reviewer #1: Yes

Reviewer #2: Yes

Reviewer #3: Yes

PLOS authors have the option to publish the peer review history of their article (what does this mean? ). If published, this will include your full peer review and any attached files.

**Do you want your identity to be public for this peer review?** For information about this choice, including consent withdrawal, please see our Privacy Policy .

Reviewer #1: No

Reviewer #2: No

Reviewer #3: No

**Figure resubmission:**
---

## [Editor Report · Decision Letter 2]

24 Feb 2026

Dear Ms Pochinok,

We are pleased to inform you that your manuscript 'iSTTC: a robust method for accurate estimation of intrinsic neural timescales from single-unit recordings' has been provisionally accepted for publication in PLOS Computational Biology.

Best regards,

Sacha Jennifer van Albada

Academic Editor

PLOS Computational Biology

Daniele Marinazzo

Section Editor

PLOS Computational Biology

---

## [Editor Report · Acceptance letter]

PCOMPBIOL-D-25-01542R2

iSTTC: a robust method for accurate estimation of intrinsic neural timescales from single-unit recordings

Dear Dr Pochinok,

I am pleased to inform you that your manuscript has been formally accepted for publication in PLOS Computational Biology. Your manuscript is now with our production department and you will be notified of the publication date in due course.

With kind regards,

Anita Estes
